# Convergent iridescence and divergent chemical signals in sympatric sister-species of Amazonian butterflies

**Joséphine Ledamoisel[1,2,3,4]\*, Bruno Buatois[5], Rémi Mauxion[3], Christine Andraud[4], Melanie McClure[6], Vincent Debat[1,2], Violaine Llaurens[1,2]**

[1]Collège de France, Centre Interdisciplinaire de Recherche en Biologie, Paris, France; [2]Muséum national d'Histoire naturelle, Institut de Systématique, Evolution et Biodiversité (ISYEB UMR 7205), Paris, France; [3]Smithsonian Tropical Research Institute, Panama, Panama; [4]Muséum national d'Histoire naturelle, Centre de Recherche sur la Conservation, Paris, France; [5]Université de Montpellier, Centre d'Ecologie Fonctionnelle et Evolutive, Montpellier, France; [6]Université de Guyane, Laboratoire Écologie, Évolution, Interactions des Systèmes Amazoniens (LEEISA), Cayenne, France

**\*For correspondence:** josephine.ledamoisel@outlook.fr

**Competing interest:** The authors declare that no competing interests exist.

## eLife Assessment

This study presents a **valuable** assessment of and **solid** evidence for increased similarity in visual appearance combined with increased chemical differences between two butterfly species in sympatry compared with differences between three populations of one of the two species in allopatry. The similarity in visual appearance hints to an evolutionary response to shared predators (but alternative explanations are possible). Thus, the difference in chemical signaling likely helps to avoid between-species mating in sympatry.

**Abstract** The evolution of traits in closely-related species living in sympatry strongly depends on both shared selective pressures and reproductive interference. In closely-related *Morpho* butterfly species living in the understory of the neo-tropical rainforest, the blue iridescent coloration of the wings is likely involved in predation evasion, as well as in mate recognition and courtship. We used spectrophotometry, behavioral experiments, visual modeling, and chemical analyses to characterize the evolution of visual and chemical traits in two closely-related species, *Morpho helenor* and *Morpho achilles*. We specifically compared trait variation between samples from allopatric and sympatric populations of *M. helenor* to test the effect of ecological interactions with *M. achilles* on trait evolution. We quantified the differences in wing iridescence and tested for variations in the sexual preference for this trait. We found a strong similarity in iridescence between *M. helenor* and *M. achilles* in sympatry, while the iridescence of *M. helenor* diverged in allopatry, suggesting that predation favors local resemblance. Although intraspecific behavioral experiments showed that iridescent signals could be used as visual cues during intraspecific mate choice, the strong resemblance of the iridescent signals between species may impair these species' visual recognition. In contrast, the divergent chemical bouquets detected between species suggest that the visual similarity of sympatric *Morpho* species may have favored the divergence of alternative traits involved in species recognition, such as chemical cues.

## Introduction

Numerous ecological interactions can impact trait evolution in closely-related species in sympatry, leading to trait convergence or divergence (*terHorst et al., 2018*). Closely-related species often display similar suites of traits because of their shared evolutionary history (*Blomberg et al., 2003*). Local selection can also act as a filter and prevent trait divergence (*Keddy, 1992*), therefore enhancing trait similarity between closely-related species when they occur in sympatry (*Chazot et al., 2014*). Trait similarity among sympatric species within a given ecological niche can thus stem from retention of locally adapted ancestral traits or from evolutionary convergence (*Muschick et al., 2012*). Selective pressures promoting the retention of locally adapted traits within species and/or trait convergence among sympatric species can also be due to local ecological interactions: for instance, shared predation pressure may promote the convergence of predator-deterrent traits in sympatry, but allow the trait to differentiate in allopatry (*Mallet, 1999*).

However, when closely-related sympatric species share a given trait, either as a result of ancestry and/or convergence, they often diverge in other traits because (1) they may be partitioned in different ecological niches (*Berlocher and Feder, 2002*), or (2) as a result of character displacement due to reproductive interference (*Grether et al., 2020*) or reinforcement due to poor hybrid fitness (*Butlin and Smadja, 2018*). As a result, traits involved in sexual competition or mate choice tend to diverge significantly more often between species in sympatry compared to allopatry (*Haavie et al., 2004*; *Marko, 2005*).

In this study, we investigate how ecological interactions in sympatry can constrain natural and sexual selection shaping trait evolution. We specifically focus on traits submitted to both natural and sexual selection and compare differences in these traits in allopatric *vs.* sympatric ranges. Theoretical and empirical studies have shown that sexual selection may favor the evolution of preferences for locally-adapted traits within species (*Servedio, 2004*; *Servedio and Boughman, 2017*; *van Doorn et al., 2009*). For instance, the predator-deterrent coloration of poison frogs is also detected and used as mating cues by females (*Reynolds and Fitzpatrick, 2007*). Similarly, habitat-dependent coloration of sympatric cichlid fish is also used as a visual cue for mate recognition (*Seehausen et al., 2008*). Yet, sexual interactions are likely to occur between individuals from closely-related species when they live in sympatry, and similar preferences for adaptive traits may thus result in substantial reproductive interference (*Gröning and Hochkirch, 2008*; *Soni et al., 2025*). Hybrids, when produced, can be unfit, thus favoring the evolution of sexual preferences for species-specific cues, rather than locally-adapted traits (*Maisonneuve et al., 2024*). To determine to what extent ecological interactions shape trait evolution, it is thus necessary to compare patterns of trait evolution in sympatry and allopatry: allopatric populations indeed allow us to estimate background levels of divergence or similarity that arise in the absence of direct ecological interactions (*Pfennig and Pfennig, 2009*). Comparing variations in adaptive traits in sympatric *vs.* allopatric populations of recently-diverged species and testing the sexual preference for those traits can shed light on the selective processes targeting traits modulating reproductive isolation and co-existence in sympatry.

In butterflies, the evolution of wing color patterns can be influenced by both natural and sexual selection. The visual discrimination of wing color pattern can enable intraspecific recognition during courtship in many species (*Costanzo and Monteiro, 2007*; *Li et al., 2017*). However, the evolution of wing color patterns is also strongly influenced by the risk of detection and/or recognition by predators (*Finkbeiner et al., 2014*; *Oliver et al., 2009*). Whether these opposite selective pressures ultimately promote trait convergence or divergence in sympatric species might depend on their relatedness: for instance, a study in *Papilionidae* showed multiple color pattern convergences between distantly-related species living in sympatry, while divergent colorations are found in closely-related species (*Puissant et al., 2023*). Divergence in traits involved in species recognition could be favored because of higher reproductive interference in closely related species than in distantly related taxa (*Pfennig and Pfennig, 2009*). In sympatric species with chemical defenses, such as *Heliconinii* butterflies, local predation pressures tend to promote the convergence of similar conspicuous warning wing patterns compared to allopatric species (i.e. Müllerian mimicry, *Joron et al., 1999*; *Merrill et al., 2014*). But the costs associated with hybrid production, in turn, favor the evolution of alternative divergent mating cues in mimetic butterflies (*Estrada and Jiggins, 2008*), and divergence in male pheromone bouquets and female attraction has been found among mimetic sister species (*González-Rojas et al., 2020*). Similarly, the evolution of specific visual mate recognition signals, limiting reproductive interference

but indistinguishable by predators, can also be promoted on the wings of mimetic butterflies (*Llaurens et al., 2014*).

Here, we focus on the evolution of mating cues in the neotropical butterfly genus *Morpho,* where multiple closely-related species co-exist in sympatry (*Blandin and Purser, 2013*). In the *Morpho* species observed in the understory, striking iridescent blue coloration is displayed on the dorsal side of the wings, due to specific wing scale structures (*Giraldo et al., 2016*; *Siddique et al., 2013*). The light signal reflected by iridescent surfaces can be very directional, as hue and brightness of iridescent objects can drastically change depending on the light environment or the observer's position (*Doucet and Meadows, 2009*). While the iridescent blue color is probably ancestral to the diversification of the understory clade (*Chazot et al., 2021*), the precise reflectance spectra at different angles likely differ among *Morpho* species. Directional iridescent signals generated in animals can likely enhance recognition by mates while remaining poorly detected by predators (*Endler, 1992*). In birds (*Simpson and McGraw, 2019*) and butterflies (*White et al., 2015*), the specific directional signal produced by the iridescent trait can be used as a cue during courtship, suggesting that the antagonistic sexual and natural selective pressures may finely tune the evolution of iridescent effects. How much sexual selection shapes the evolution of iridescent properties in sympatric *Morpho* species is currently unknown, but behavioral experiments carried out in the field in Amazonian Peru highlighted strong visually-based territorial interactions among males from sympatric species and limited species discrimination based on female coloration in males (*Le Roy et al., 2021b*).

This raises questions on the key visual cues involved in mate choice, given that the iridescent blue coloration shared by closely-related species encountered within the understory is also likely under selection by predators. The iridescent bright blue dorsal coloration of *Morpho* wings contrasts with the brown and matte ventral side and generates a peculiar visual effect during flight. The combination of the blue flashes produced by the alternate exposure of the bright blue *vs.* brown sides of the wings during flapping flight, in addition to erratic flight trajectories, makes these *Morpho* very difficult to catch by bird predators (*Young, 1971*), potentially enhancing their evasive capabilities (*Murali and Kodandaramaiah, 2020*). Experimental trials with evasive prey have shown that predators learn to avoid prey they repeatedly fail to catch (*Páez et al., 2021*). The display of iridescent wings could thus be associated with a higher survival rate in nature because of both (i) direct effects, through successful escape of predator attacks, and (ii) indirect effects, by limiting predation attempts by birds recognizing the blue signals and refraining from attacking, as highlighted by butterfly release experiments investigating the hunting behavior of wild insectivorous birds in Brazil (*Pinheiro et al., 2016*). Mark recapture experiments in the field with manipulated dorsoventral contrasts in wild Morphos have suggested that dynamic flash coloration can reduce predation rate (*Vieira-Silva et al., 2024*). This indirect effect could promote the evolution of convergent blue patterns in sympatric species, similar to the mimicry observed in species with chemical defenses (*Joron et al., 1999*; *Merrill et al., 2014*). In line with this hypothesis, repeated local convergence in the proportion of iridescent blue *vs.* black areas on the dorsal side of the wings has been documented in the sister-species *Morpho helenor* and *Morpho achilles* living in sympatry throughout the Amazonian basin (*Llaurens et al., 2021*). Precise quantification of variations in iridescence is now needed to assess the respective effects of selection by predators and mates that may drive convergent *vs.* divergent evolution of iridescence in sympatric and allopatric ranges.

First, we quantified iridescence in allopatric vs. sympatric populations of *M. helenor* subspecies. Since coloration is expected to be more similar within than among species under neutral evolution, we used allopatric populations of *M. helenor* as a baseline to assess convergence of iridescence between two sympatric species (*M. helenor* and *M. achilles*). We then conducted behavioral experiments to test the effect of variation in iridescence on mate recognition, using two subspecies of *M. helenor* displaying different iridescent phenotypes. This intraspecific comparison allows identifying the visual cues used in mate choice in *M. helenor,* teasing apart the effects of iridescence and/or wing pattern. We then tested whether those visual cues are used in species recognition between sympatric *M. helenor* and *M. achilles*. Finally, we studied variations in the volatile compounds produced by wild males and females from sympatric populations of the two species to explore the evolution of potentially alternative traits, such as chemical cues, possibly acting as a reproductive barrier.

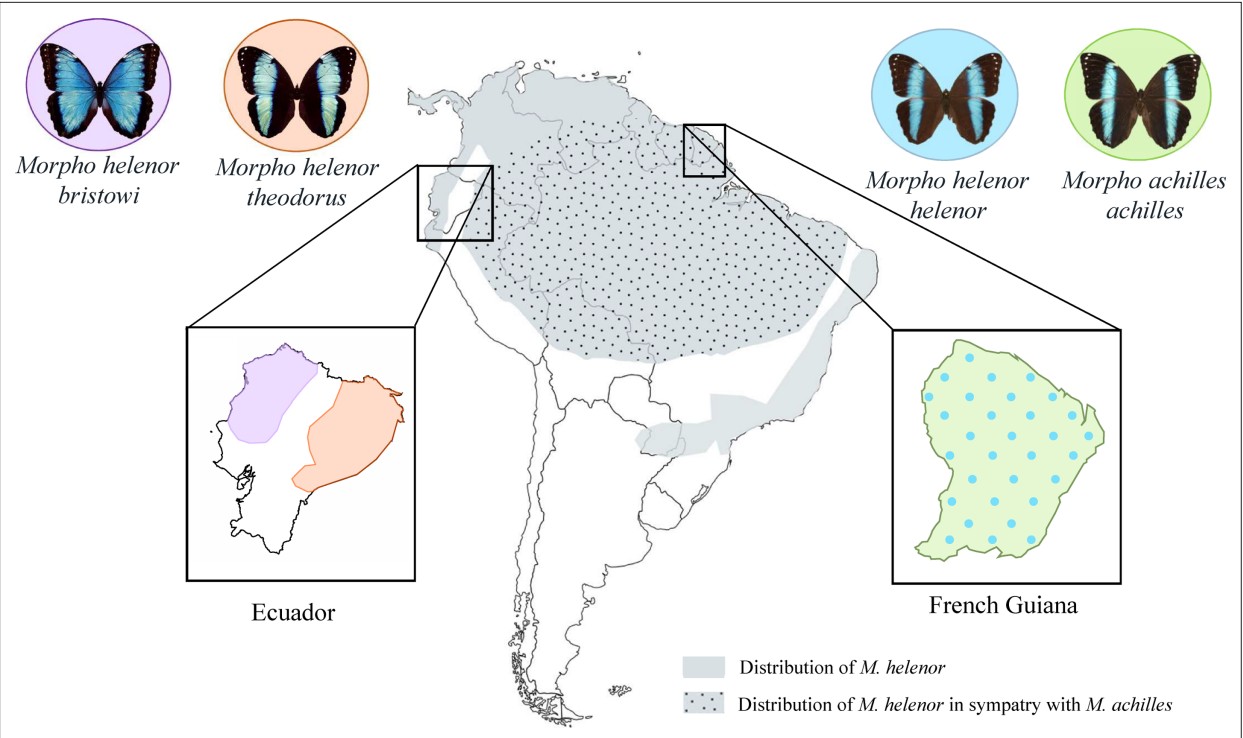

**Figure 1.** Distribution of the sister-species *M. helenor* and *M. achilles* across South America. The gray area represents the whole distribution of *M. helenor*, and the dotted area indicates the localities where *M. helenor* is in sympatry with *M. achilles*. Note that the different subspecies of *M. helenor* are found in different localities and display substantial variations in the proportion of black *vs.* blue areas on the dorsal sides of their wings. For example, *M. h. bristowi* and *M. h. theodorus* are allopatric in Ecuador, with their respective distribution separated by the Andes such that they are never in contact. While *M. h. bristowi* is found on the Pacific side of the country and has a wide blue band on the dorsal side of its wings, *M. h. theodorus* is found in Western Amazonia and displays a narrow blue band on the dorsal side of its wings. However, *M. helenor* is sympatric with *M. achilles* throughout the Amazonian rainforest. In French Guiana, the subspecies *M. helenor helenor* and *M. achilles achilles* display convergent dorsal color patterns with thin blue bands (***Llaurens et al., 2021***).

## Results

We investigated differences in wing iridescence, color perception, and color pattern-based mate choice of two sister species of *Morpho*, *M. helenor* and *M. achilles*, which diverged ~3.6 Mya (***Chazot et al., 2021***). We relied on geographic variations of wing color patterns within *M. helenor* (***Figure 1***) to compare the iridescence of three different populations of *M. helenor* (subspecies *M. helenor bristowi*, *M. helenor theodorus,* and *M. helenor helenor*) and test the perception of these patterns by predators and mates using visual models.

### The convergent iridescent signals of sympatric Morpho species are not differentiable

We measured the reflectance of *Morpho* wings at different angles of observation and illumination, in two different planes (referred to as proximo-distal and antero-posterior planes) following two complementary protocols (referred to as 'specular' and 'tilt'; see ***Figure 2A*** and Appendix 1 for extended methods). We then extracted for each spectrum three commonly used colorimetric variables: Hue, Brightness, and Chroma (analyzed in ***Figure 2—figure supplement 1***).

We found that hue and brightness always vary depending on the angle of incidence of the light, as confirmed by PERMANOVAs detecting a significant effect of the 'Angle' variable for all methods of measurements (***Figure 2***; statistical analyses in ***Appendix 2—table 1*** and ***Figure 2—figure supplement 2***) confirming that in both species, both sexes are iridescent. Second, the brightness of *Morpho* wings is highly sexually dimorphic, as the effect of sex was always significant. The brightness measured with the 'tilt' protocol (***Figure 2F and G***) showed that males tend to be brighter than females in all populations.

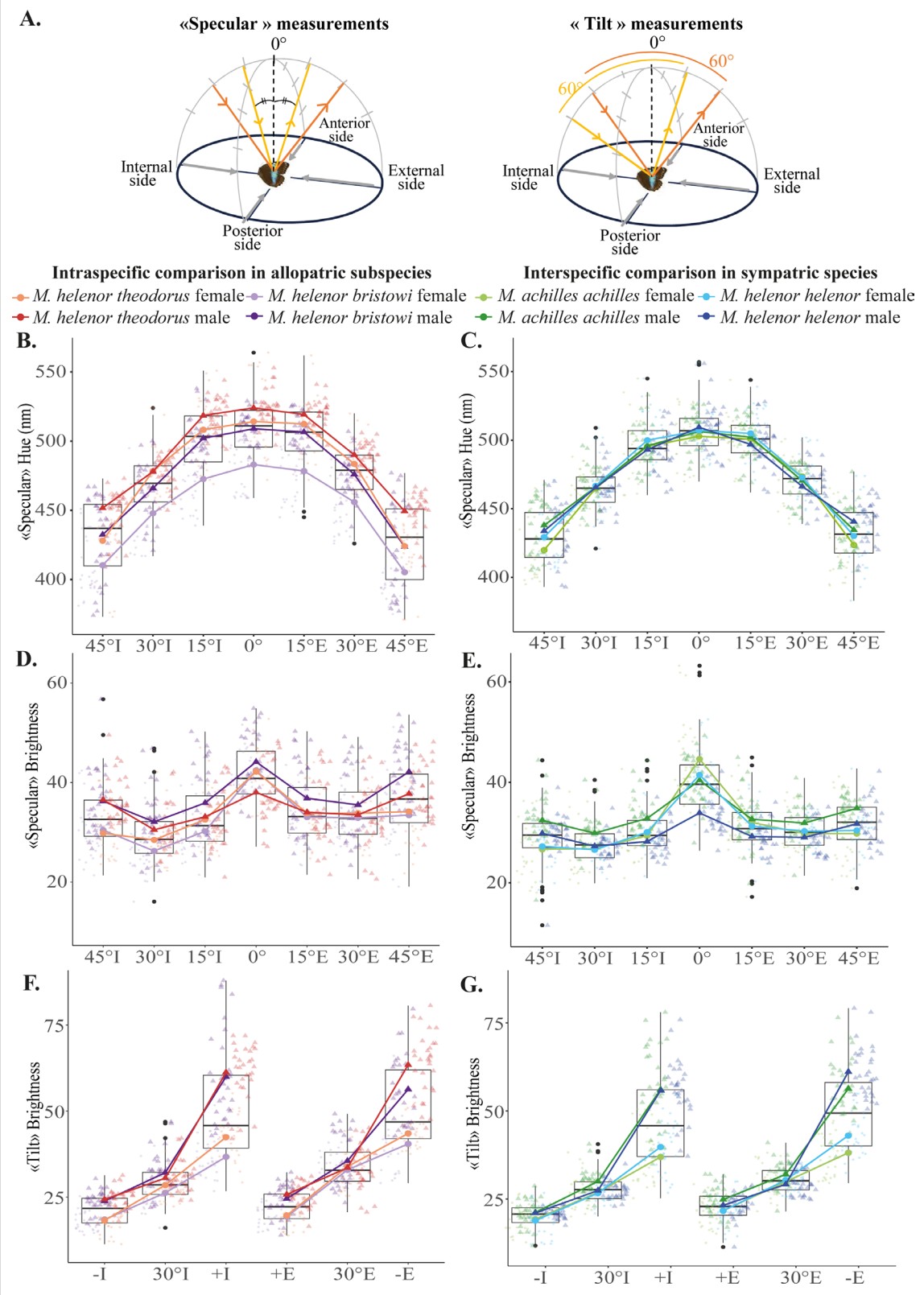

**Figure 2.** Differences in hue and brightness on the proximo-distal plane of *Morpho* wings. (**A**) Illustration of the two protocols used to assess the differences in *Morpho* wing reflectance. The 'Specular' set-up allows for the quantification of the wing color variations, while the 'tilt' set-up can be used to quantify brightness variation for each *Morpho* wing (see Appendix 1 for extended methods). Variations of hue (**B** and **C**) and brightness (**D** and **E**) calculated from the wing reflectance measured with the 'Specular' set-up, and variations of brightness calculated from the wing reflectance measured

*Figure 2 continued on next page*

*Figure 2 continued*

from the 'tilt' set-up (**F** and **G**). Those hue and brightness parameters were calculated for the allopatric *M. h. theodorus* and *M. h. bristowi* (first column in orange and purple) and for the sympatric *M. h. helenor* and *M. a. achilles* (second column in green and blue) on the proximal-plane plane of their wings (I=illumination on the internal side of the wings, E=illumination on the external side of the wings). See *Appendix 2—table 1* for the PERMANOVA analyses describing those graphs.

The online version of this article includes the following figure supplement(s) for figure 2:

**Figure supplement 1.** Differences in chroma on the proximo-distal and anteroposterior plane of *Morpho* wings.

**Figure supplement 2.** Differences in hue and brightness on the anteroposterior plane of *Morpho* wings.

To finely appraise the complete spectrum of variation in iridescence within and among *Morpho* species, we then used a multivariate approach allowing us to analyze the full reflectance spectra obtained from both proximo-distal and anteroposterior measurements. At the intraspecific level, significant divergence of iridescence was detected between allopatric subspecies of *M. helenor* (*Figure 3A*). In contrast, iridescence did not differ between species living in sympatry (PERMANOVA, $F=2.09$, df = 1, p-value = 0.084), suggesting that iridescence converges in sympatry (*Figure 3B*). In both cases, the interaction term taxon*sex was non-significant, suggesting that the sexual dimorphism of iridescence was similar for all taxa.

We then modeled the perception of color contrast in the visual systems of *M. helenor* and UV-sensitive birds respectively, and tested whether the differences of reflectance observed between sexes and species at different angles of illumination could potentially be perceived by a *Morpho* observer or a putative predator. As sexual dimorphism of wing reflectance was detected, we measured the achromatic distances and chromatic distances for males and females separately. Achromatic distance results (*Figure 3—figure supplement 1*) suggest that neither birds nor *Morpho* butterflies could theoretically differentiate the brightness of the wings of sympatric or allopatric individuals, based on visual modeling. Concerning chromatic distances, as the results were similar along the two directions of measurements (antero-posterior and proximo-distal), only proximo-distal results are shown in *Figure 3C* (antero-posterior results are shown in *Figure 3—figure supplement 2*). Results suggest that *Morpho* butterflies and predators can theoretically visually perceive the difference in the blue coloration between the different subspecies of *M. helenor*: mean chromatic contrasts were always higher than 1 JND, for both sexes and for all angles, using both bird and *Morpho* visual models. However, this effect was stronger for females. In males, confidence intervals were only strictly significant for the 0° illumination using the *Morpho* visual model, suggesting a weak discrimination at most angles of observation.

In contrast, the mean chromatic contrasts measured between sympatric species were always below 1 JND: the minute differences observed between sympatric species are thus unlikely to be discriminated either by *Morpho* or bird predators.

## Evidence of mate preferences based on visual cues within species

Experimental evidence has shown that *M. helenor* males are attracted to blue coloration when patrolling in the wild (*Le Roy et al., 2021b*). We thus performed a series of mate choice experiments to determine the nature of the visual cues used in mate choice in *M. helenor*. We focused on the two previously studied allopatric subspecies of *M. helenor* (*M. h. bristowi* and *M. h. theodorus*), which significantly differ in both wing pattern and iridescence.

We first tested whether mating tended to be preferentially assortative by conducting a tetrad experiment involving one male and one female of both subspecies. The highest number of mating events occurred between the two assortative pairs (see *Table 1*). Indeed, a Fisher's exact test performed on the contingency table showed a significant departure from random mating (p-*value = 0.04*). The time until an assortative mating (mean = 48 min, sd = 38 min) or until a disassortative mating (mean = 48 min, sd = 38 min), however, did not significantly differ (*T-test; stat = –0.27*, p-*value = 0.79*). Note that the effects were clearly driven by the bolder behavior of *M. h. bristowi* individuals, as half of the mating events occurred between a female and a male of this subspecies.

To test whether this assortative mating was driven by visual cues and to identify these cues, we performed a series of experiments using dummy butterflies treated with hexane to remove any olfactory cue. We specifically focused on studying allopatric *M. h. theodorus* and *M. h. bristowi* because

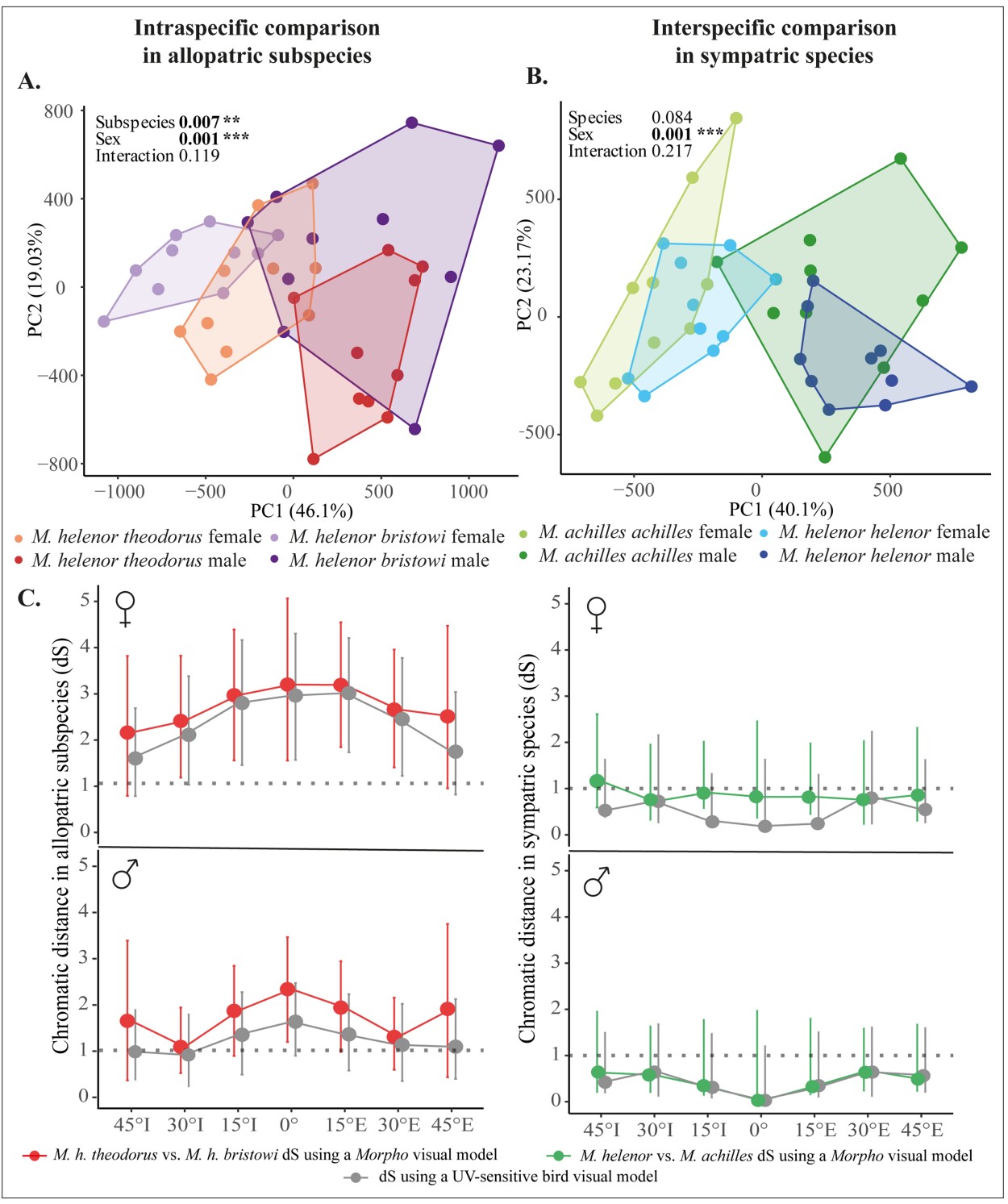

**Figure 3.** Characterization and perception of the iridescent coloration of *Morpho* butterflies. (**A**) and (**B**): PCAs showing the variation in iridescence for both sexes, (**A**) in the two sympatric species from French Guiana (*M. h. helenor* vs. *M. a. achilles*) and (**B**) in the two allopatric Ecuadorian subspecies of *M. helenor* (*M. h. theodorus* vs. *M. h. bristowi*). Each point represents the global signal of iridescence of each individual, corresponding to the 21 complete reflectance spectra obtained from the 21 tested angles of illumination. The results of the PERMANOVA are shown on the top left corner of each graph. (**C**) The chromatic distances (i.e. the visual discrimination rate by a visual model) of the wing reflectance measured with the 'Specular' set-up on the proximo-distal plane. Visual modeling was used to calculate the chromatic contrast of blue coloration between allopatric *M. helenor* subspecies (red) and between the two sympatric sister-species *M. helenor* and *M. achilles* (green), as perceived by a *Morpho* visual system for every angle of illumination measured on the proximo-distal plane. The chromatic contrast likely perceived by UV-sensitive birds is shown in gray. Chromatic contrast

*Figure 3 continued on next page*

*Figure 3 continued*

of the female wings (top) and male wings (bottom). The threshold of discrimination is shown by the dotted line and set to 1 *Just Noticeable Difference* (*JND*). Error bars show the confidence intervals calculated during the bootstrap analysis.

The online version of this article includes the following figure supplement(s) for figure 3:

**Figure supplement 1.** Achromatic distances of the wing reflectance measured with the 'Specular' set-up on the anteroposterior and proximodistal plane.

**Figure supplement 2.** Chromatic distances of the wing reflectance measured with the 'Specular' set-up on the anteroposterior plane.

they both differ in their iridescent signal and in their iridescent pattern. Different pairs of female dummies were presented to *M. helenor* males to investigate assortative preference between subspecies (experiment 1), discrimination of the wing pattern (experiment 2), and discrimination of the iridescent coloration (experiment 3; see the experimental details in the Materials and methods section).

*Experiment 1* tested male assortative visual preference using two dummies each displaying the wings of a female of a different *M. helenor* subspecies. *M. h. theodorus* males did not show any preference for any female dummy, as shown by the probability of approaches and touches that were not different from the random 0.5 expectation (Binomial GLMM approaches: Estimate = −0.121 ± 0.091 SE, z=−1.332, p-*value = 0.183*, **Figure 4A**; touches: Estimate = 1.600 ± 1.094 SE, z=1.462, p-*value = 0.144*, **Figure 4B**). *M. h. bristowi* males, in contrast, approached and touched a con-subspecific *M. h. bristowi* dummy female significantly more often than expected by chance alone (Binomial GLMM approaches: Estimate = 0.430 ± 0.105 SE, z=4.11, p-value <0.0001, **Figure 4A**; touches: Estimate = 1.646 ± 0.273 SE, z=6.029, p-value <0.0001, **Figure 4B**), suggesting a preference toward the *M. h. bristowi* females based on the recognition of visual cues alone.

*Experiment 2* tested male pattern-based discrimination, using two dummies made with the wings of *M. h. bristowi* females (i.e. same iridescence), but one of which was modified so as to present the thin blue bands found in *M. h. theodorus* (i.e. a different pattern). *M. h. bristowi* males tended to approach and touch significantly more often the wild-type *M. h. bristowi* female model as compared to the modified *M. h. bristowi* female model exhibiting a *theodorus* pattern (Binomial GLMMs approaches; Estimate = 0.599 ± 0.089 SE, z=6.751, p-value <0.0001, touches; Estimate = 2.805 ± 0.944 SE, z=2.972, p-value = 0.003, see **Figure 4—figure supplement 1**). As both female dummies shared the same blue iridescence, the choice of *M. h. bristowi* males was likely influenced by the difference in the pattern displayed by the dummy females, suggesting that the color pattern is used by males during mate choice.

*Experiment 3* tested male color-based mate discrimination using two dummies made with the wings of females from the two subspecies (i.e. different iridescence), but those of *M. h. bristowi* were modified so as to resemble *M. h. theodorus* (i.e. same pattern). *M. h. theodorus* males tended to approach and touch significantly more often the *M. h. theodorus* female wild-type model compared to the artificially modified, theodorus-like *M. h. bristowi* female dummy (Binomial GLMMs on approaches; Estimate = 0.348 ± 0.121 SE, z=2.867, p-value = 0.004, on touches; Estimate = 1.172 ± 0.465 SE, z=2.521, p-value = 0.012, see **Figure 4—figure supplement 2**). As both dummy females shared the same color pattern, the preference expressed by *M. h. theodorus* males likely stemmed from the differences in the iridescent blue color, suggesting color-based discrimination in males.

In experiments 2 and 3, modified dummies were each time strongly rejected by males. Although all dummies contained traces of black marker to control for manipulation and treatment, we cannot entirely rule out that the artificial modifications performed on dummies could generate a specific visual rejection by males, because we cannot be sure whether butterflies perceive these modifications

**Table 1.** Tetrad experiment results.

Number of mating events involving the different possible pairs of males and females from the two subspecies of *M. helenor* (*bristowi* vs. *theodorus*) in 30 tetrad experiments.

|  | *M. h. theodorus* female | *M. h. bristowi* female |
|---|---|---|
| *M. h. theodorus* male | 6 | 3 |
| *M. h. bristowi* male | 5 | 16 |

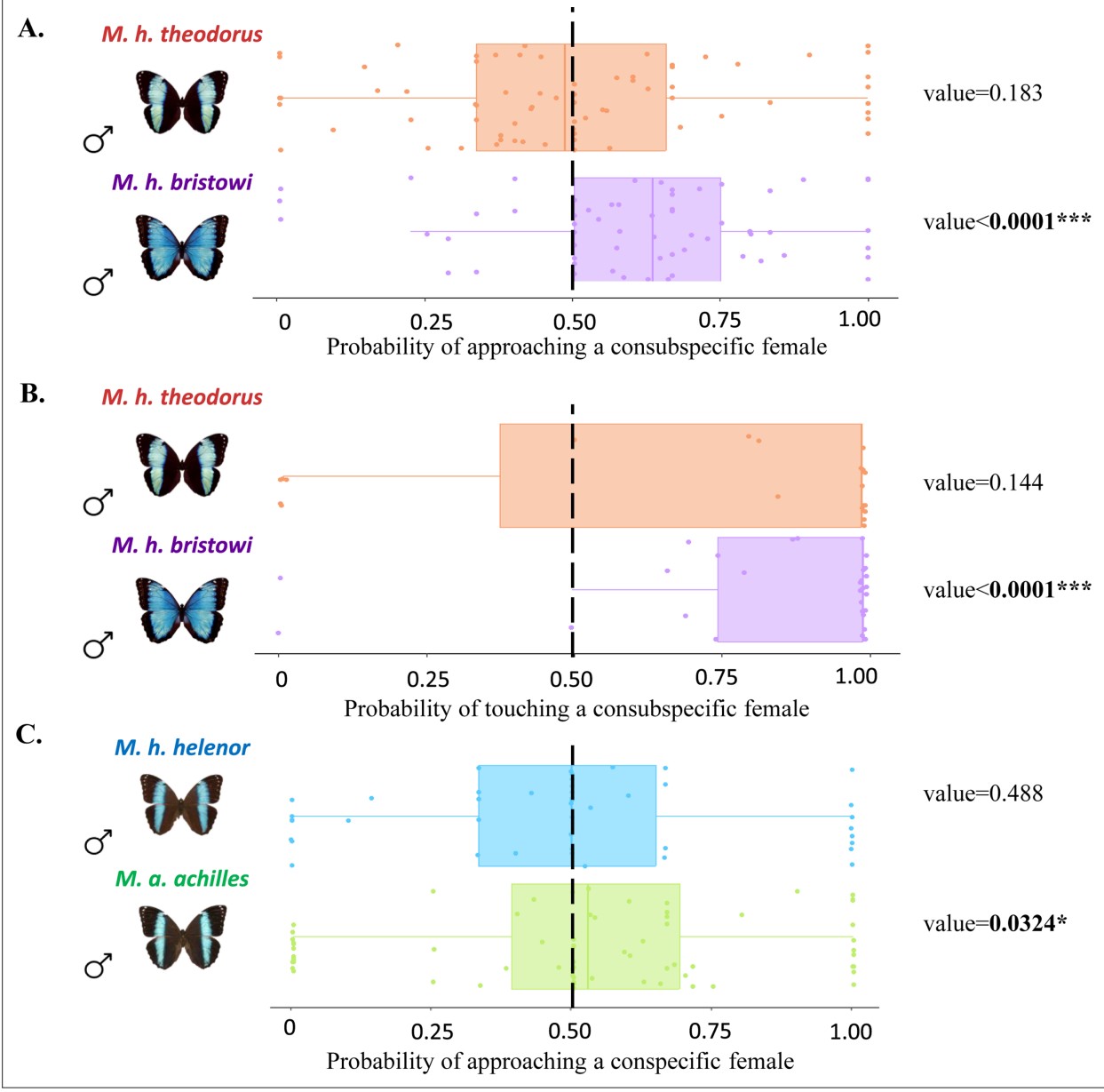

**Figure 4.** *Morpho* male preference based on visual cues alone. Probabilities of (**A**) approaching and (**B**) touching a con-subspecific female for *M. h. bristowi* males (purple) and *M. h. theodorus* males (orange) from Ecuador, as well as the probabilities of (**C**) approaching a con-specific female for *M. h. helenor* males (blue) and *M. a. achilles* males (green) from French Guiana, measured during *experiment* 4. The dotted line indicates the expected probability of approaching/touching a model if no preference is present. The *p-values* resulting from the binomial GLMM testing for preferentially directed interactions toward one female dummy are shown next to the corresponding graphs.

The online version of this article includes the following figure supplement(s) for figure 4:

**Figure supplement 1.** *Morpho* male pattern-based discrimination (experiment 2).

**Figure supplement 2.** *Morpho* male color-based discrimination (experiment 3).

as equivalent to natural coloration. These experiments nevertheless suggest that visual cues – color pattern and iridescence – jointly contribute to mate choice in *M. helenor*.

### Visual cues are not pre-zygotic reproductive barriers between the mimetic species M. h. helenor and M. a. achilles

We then tested whether these visual cues allowed males to express preferences toward their own species in the sympatric range of *Morpho achilles* and *M. helenor*. In a final experiment (experiment 4), we thus investigated whether males from the two species *M. achilles* and *M. helenor*, sampled in their sympatric range (French Guiana), have a visual preference toward their conspecific females: two female dummies made with the wings of females from the two species were presented to each male. While none of the males (neither from *M. helenor* nor from *M. achilles*) touched the female models, they did approach them in the cages. The approach rate and the number of approaches did not significantly differ between *M. helenor* (rate = 61%; mean = 4.21, sd = 5.34) and *M. achilles* males (rate = 66%; mean = 7.18, sd = 8.29; *binomial GLMM: p-value = 0.626*), revealing similar levels of male courtship motivation. The independent analysis of the preference per species (*Figure 4C*) shows that *M. helenor* males did not significantly approach their conspecific female more often than the hetero-specific one (Binomial GLMM, Estimate = −0.161 ± 0.198 SE, z=−0.813, p-value = 0.416), while *M. achilles* males did (Binomial GLMM, Estimate = 0.196 ± 0.092 SE, z=2.14, p-value = 0.032). However, note that the probability for *M. achilles* to approach one of its conspecifics is only 0.54, which is very low compared to the probability of *M. h. bristowi* to approach (0.61) or touch (0.84) its consubspe-cific in experiment 1. Visual discrimination thus appears to be lower among species, between the sympatric *M. helenor* and *M. achilles*, than within species, between the allopatric *M. helenor* subspe-cies, probably due to the remarkable similarity in color pattern and iridescence between the two species in their sympatric range.

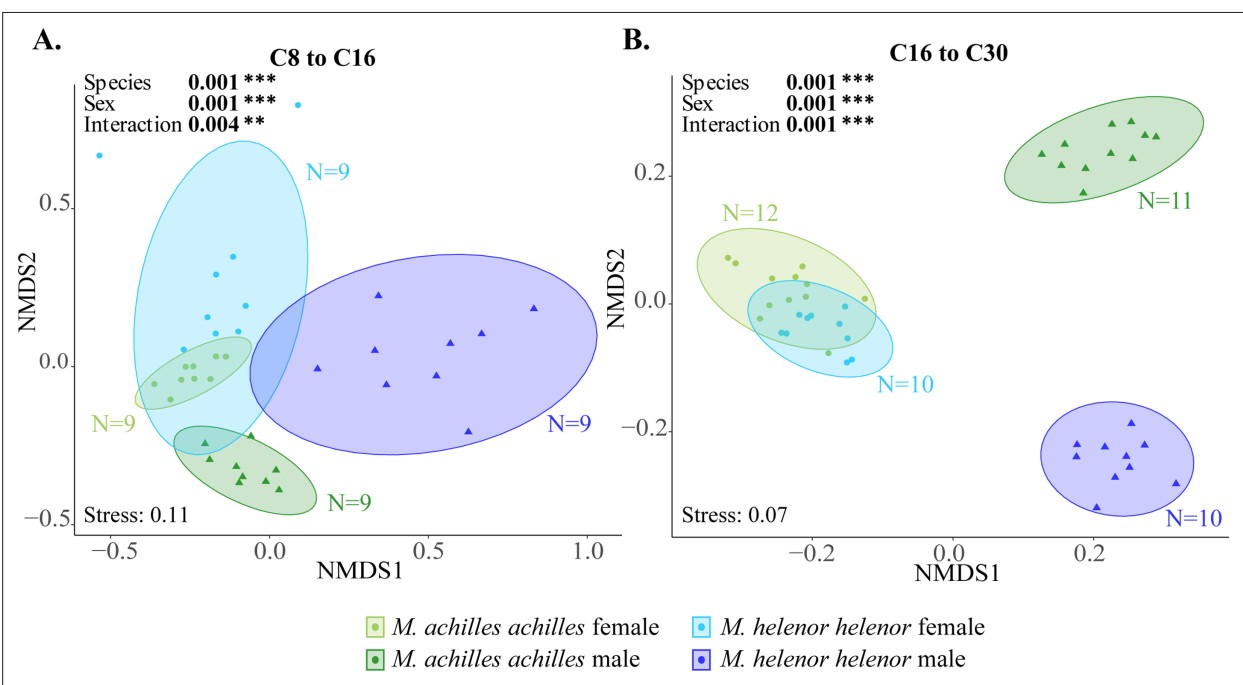

**Figure 5.** Divergent chemical compounds found on the genitalia of sympatric *Morpho* males. nMDS representation of the differences in the concentration of (**A**) C8 to C16 chemical compounds and (**B**) C16 to C30 chemical compounds found in the genitalia of *M. h. helenor* and *M. a. achilles* males and females, calculated using Bray-Curtis distances. *M. h. helenor* are shown in blue (males are in dark blue and females in light blue), and *M. a. achilles* are shown in green (males are dark green and females are light green). The result of the PERMANOVA (999 permutations) testing the effect of sex and species on the chemical composition of *M. h. helenor* and *M. a. achilles* is shown on each figure.

The online version of this article includes the following figure supplement(s) for figure 5:

**Figure supplement 1.** Chromatograms ilustrating the chemical compounds found on male and female genitalia.

### Divergent chemical profiles observed between males of the sympatric species M. helenor and M. achilles

We finally investigated the volatile chemical compounds extracted from the genitalia of males and females from the mimetic sympatric species *M. helenor* and *M. achilles*. The nMDS analyses (*Figure 5*) suggested a strong divergence in the chemical profiles between males of the two species, especially for C16 to C30 compounds. This result contrasts with the strong similarity of chemical profiles found between the females of these two species in these longer-chain compounds (*Figure 5—figure supplement 1*).

The PERMANOVA of the chemical spectra of *M. h. helenor* and *M. a. achilles* genitalia revealed a significant effect of sex, species (p-*value <0.001* on both the C8 to C16 and the C16 to C30 datasets, see *Figure 5*) and a significant interaction between the two variables. When analyzing the results for each sex separately (post-hoc), we found a significant effect of the species variable (*PERMANOVA, df = 1, F=10.013*, p-*value = 0.001* for males and *df = 1, F=1.928*, p-*value* = 0.049 for females in the C8 to C16 compounds, *PERMANOVA, df = 1, F=26.031*, p-*value = 0.001* for males and *df = 1, F=2.776*, p-*value = 0.048* for females in the C16 to C30 compounds), suggesting a substantial divergence between species in chemical bouquets, especially marked in males.

Next, we performed an Indicator Value Analysis to identify the different compounds significantly associated with each species, separating males and females (*Appendix 3—table 1*). Among all the annotated compounds, we found traces of beta-ocimene, previously identified as a butterfly pheromone. Interestingly, different proportions of this compound were found on the genitalia of *M. heleno*r and *M. achilles* males (*Appendix 3—table 1*).

## Discussion

### Convergent iridescence between sympatric Morpho butterflies suggests a prominent role of predation on the evolution of iridescence

Considering divergence time, wing iridescence is expected to be more different between the species *M. achilles* and *M. helenor* than among populations of *M. helenor*. Our results on the variation of iridescent properties of the blue patches of *M. helenor* and *M. achilles* show the opposite trend, with more divergence between populations within *M. helenor* than between *M. helenor* and *M. achilles* in their overlapping range. Together with the convergent evolution of the blue band width previously detected in multiple sympatric locations in these two species (*Llaurens et al., 2021*), this observation is consistent with a primary role of natural selection on the evolution of wing coloration. Sympatric species are indeed submitted to the same abiotic conditions, favoring the evolution of similar adaptations (*Chazot et al., 2014*). For instance, wing coloration is involved in thermoregulation in butterflies, and similar environmental conditions may constrain the evolution of this trait. Despite this, thermal absorbance in iridescent species appears independent of environmental temperature, as experimentally measured in a large sample of Lepidopteran species (*Bosi et al., 2008*). Similarly, recent experiments have shown that iridescent *Morpho* species are not better at thermoregulating than non-iridescent ones (*Bouinier et al., 2025*), suggesting that the evolution of structural colors is not driven by selection on thermoregulation. Instead, the results of the visual models showing that these convergent iridescent signals are likely undistinguishable by birds support an effect of selection by predators (*Chouteau et al., 2016*; *Stuckert et al., 2014*). The lack of directionality for the iridescent signal of the wings further supports a limited effect of sexual selection driving divergence in visual signal. Directionality in iridescent signals is indeed expected when sexually selected traits incur a predation cost due to their conspicuousness: directionality might enhance signaling to mates while remaining poorly detected by predators (*Endler, 1992*). Our study shows that *M. helenor* and *M. achilles* wings display a gradual variation of hue and brightness when changing the illumination/observation angles contrary to other iridescent butterfly species with more directional changes (e.g. *Hypolimnas bolina*; *White et al., 2015*). Recent predation experiments showed that the gradual change of hue and brightness of iridescence can be involved in crypsis and camouflage: for instance, iridescent beetles are less detected by predators when placed against a leafy background (*Kjernsmo et al., 2020*; *Thomas et al., 2023*). However, like most Satyrinae, *Morpho* butterflies tend to rest on leaves with their wings closed, thus hiding the iridescent patterns on the dorsal side of their wings at rest. Furthermore, iridescent species have a very erratic flapping flight, creating very conspicuous flashes

(*Le Roy et al., 2021a*; *Young, 1971*). Gradual variations of hue and brightness during wing movements are thus consistent with the emission of a confusing visual signal that increases the chances of escaping predators. The convergence of visual signals could have been favored because of the indirect advantage that results in a reduction in attacks from bird predators that have experienced and learned to avoid the escape capabilities of these prey (*Pinheiro and Freitas, 2014*; *Ruxton et al., 2004*). Coupled with results from predation experiments supporting an effect of dynamic flash coloration as a protective mechanism in *Morpho helenor* (*Vieira-Silva et al., 2024*), convergence of color patterns in those sympatric butterflies is consistent with the hypothesis of escape mimicry, although this warrants further research.

Interestingly, our reflectance measurements showed that males are generally brighter than females in both species, implying that the flashes generated by males during flight could be more conspicuous. Field experiments have highlighted the territorial behavior of males found patrolling along rivers and displaying competitive interactions with other males (*Le Roy et al., 2021a*). This flight behavior strongly differs from female behavior, which is much less frequently observed in the wild and typically spends more time hidden in the understory (*Young, 1973*). Considering that these ecological niche differences between males and females could result in different predation pressures in contrasted light conditions (open river banks *vs.* cluttered understory), iridescent signals could have evolved differently in the two sexes, and similarly so in both species (*Allen et al., 2011*; *Reimchen and Nosil, 2004*).

Although we only compared one sympatric population and one allopatric population, our results on geographic variations in iridescence within *M. helenor* and similarity of iridescent coloration between *M. helenor* and *M. achilles* in sympatry are consistent with an effect of local selection exerted by predators. Further investigation on multiple populations is needed to test whether this trend can be generalized to multiple sympatric locations where other wing patterns can be observed. Additionally, characterizing convergence of iridescence is an indirect method to test for the effect of iridescence in predator deterrence: testing the predation rate of *Morpho* when presented with local or exotic predators would better characterize the direct effect of convergent iridescent in escaping predators in sympatry.

### Iridescent color patterns can be used as mate recognition cues in *M. helenor*

Our tetrad and male mate choice experiments in *M. helenor* butterflies clearly show assortative mating based on wing coloration, mainly driven by the behavior of individuals from the subspecies *M. h. bristowi*. This result is consistent with the high attraction generated by blue coloration on males, as well as the preference of males for local *vs.* exotic wing color patterns found in previous field behavioral experiments carried out in Amazonian Peru (*Le Roy et al., 2021a*). Our male mate choice experiments suggest that both the wing patterning and the iridescent properties of the blue patch could have an effect on male preference. Furthermore, visual modeling also suggests that the differences in iridescent coloration between the populations of the same species can be perceived by mates. Coloration and color perception are key in mate recognition in many taxa (marine invertebrates; *Baldwin and Johnsen, 2009*, birds; *Caro et al., 2021*, mammals; *Waitt et al., 2003*) but patterns can also inform mate choice (*Houde, 1987*; *Pérez-Rodríguez et al., 2017*). In jumping spiders, both pattern and coloration are important cues used during mate choice (*Zhou et al., 2021*). Similarly, in *Heliconius* butterflies, both pigmentary coloration and wing pattern are used in mate choice, although the preference is hierarchical and coloration is the most important cue (*Finkbeiner et al., 2014*). Further experiments are needed to disentangle the relative effects of iridescent coloration and wing pattern in mate preference in *Morpho* butterflies, but our results show that iridescent color patterns can be used as mate recognition cues in *M. helenor*.

### Evolution of visual and olfactory cues in similar sister-species living in sympatry

Interestingly, intraspecific variations in courtship motivation were detected in our behavioral experiments within the species *M. helenor*. Individuals from the western areas of the Andes displayed high courtship motivation while butterflies from the Amazonian area, where *M. helenor* coexists with *M. achilles,* showed a more limited response to our visual dummies. The joint convergence of both

color pattern and blue iridescence between *M. helenor* and *M. achilles* in Amazonia probably makes visual cues poor species recognition signals: visual modeling even suggests that *M. helenor* cannot perceive the color difference between the two, whereas it could make the difference between two *M. helenor* subspecies displaying more divergent coloration. This is further supported by our male choice experiment showing that although *M. helenor* males living in sympatry with *M. achilles* are attracted by blue iridescent colors and approach each female, they do not preferentially interact with their conspecific female dummies. Nevertheless, *M. achilles* males tended to approach significantly more of their conspecific females. Previous mate choice experiments performed in the wild suggested that *M. achilles* indeed preferred to fly around *M. achilles* females compared to *M. helenor* females (*Le Roy et al., 2021a*), suggesting that visual cues are still relevant in mate choice for some *Morpho* species living in sympatry with other *Morpho* butterflies. However, in our experiments, *M. achilles* males approached their conspecifics with a probability of 0.54 only, which is very low compared to what has been observed between intraspecific *M. helenor* subspecies, probably because of the high resemblance between *M. helenor* and *M. achilles* in sympatry.

The convergent evolution of iridescence in sympatric species might thus promote divergent evolution of alternative mating cues. By quantifying chemical compounds found on the genitalia of males and females sampled in Amazonian French Guiana, we did indeed find a strong divergence in the chemical profiles of sympatric *M. helenor* and *M. achilles,* especially in males. High male chemical divergence in sympatric species suggests the evolution of female mate choice based on olfactory discrimination (*Mérot et al., 2015*). Although all compounds were not identified, some of those that were have previously been identified as pheromone compounds used during male/female interactions in butterflies. In particular, beta-ocimene was found on the genitalia of males and is known to contribute to mate choice in courtship (*Li et al., 2017*) or as an anti-aphrodisiac used by *Heliconius* males to repel other males after mating with a female (*Schulz et al., 2008*). Additionally, we found strong divergence between males for heavier chemical compounds (C16-C30). Because cuticular hydrocarbons can be involved in mate discrimination during courtship (*Ômura et al., 2020*), diverging non-volatile compounds could also be used as a discriminating cue. Although we did not perform mate choices to test for the discrimination of olfactory cue by females, divergence of male odorant cues could be due to reproductive interference between sympatric species (*Bacquet et al., 2015*; *Dyer et al., 2014*). Altogether, our results therefore suggest that the convergent evolution of iridescent wing pattern between the sympatric *M. helenor* and *M. achilles* in Amazonia, likely promoted by shared predation pressures, may have negatively impacted visual discrimination and favored the evolution of divergent olfactory cues. However, determining whether divergence of the chemical profiles of allopatric and visually more divergent *Morpho* butterflies occurs and to what extent would be needed in order to determine whether the divergence observed in sympatric *Morpho* species is truly due to reinforcement.

## Conclusions

We found evidence of convergent iridescent patterns in sympatry suggesting that predation could play a major role in the evolution of iridescence. Further work is nevertheless needed to directly test this hypothesis and establish the importance of evasive mimicry in *Morpho* (*Pinheiro et al., 2016*), similar to the Müllerian mimicry observed in chemically-defended species. There are, however, striking differences between the two types of mimicry: while a wide diversity of conspicuous color patterns is involved in Müllerian mimicry (*Mallet and Gilbert, 1995*), the variation of iridescent patterns involved in evasive mimicry might be more tightly constrained, because iridescence itself directly participates in survival via visual confusion (*Murali and Kodandaramaiah, 2020*). Moreover, a switch in color pattern is generally considered a magic trait facilitating speciation in chemically defended species (*Merrill et al., 2012*), and Müllerian mimicry frequently occurs among distantly-related species (*Puissant et al., 2023*). Here we observe the co-existence of Morpho sister-species displaying convergent color patterns, suggesting that speciation should be promoted by the evolution of traits other than visual, such as olfactory chemical cues, or by the specialization into divergent ecological temporal niches (*Le Roy et al., 2021a*). Altogether, this study addresses how convergence in one trait as a result of biotic interactions may alter selection on traits in other sensory modalities, resulting in a complex mosaic of biodiversity.

# Materials and methods

## Study system

We investigated trait variations in the sister-species *M. achilles* and *M. helenor* that occur in sympatry in the Amazonian basin (*Figure 1*), with two main sampling schemes, focusing on intraspecific and interspecific variations, respectively.

### M. helenor populations of Ecuador

Pupae from the Ecuadorian populations of *M. h. bristowi* and *M. h. theodorus* were purchased from a breeding farm located in Ecuador (Quinta De Goulaine, https://quintadegoulaine.com/es/papillons. php), frequently supplying their breeding with wild-caught individuals from Tena (Eastern Ecuador) and Pedro Vicente Maldonado (Western Ecuador) respectively. No artificial selection was performed. These commercially-bought pupae were then raised in insectaries at STRI in Gamboa, Panama, between January and March 2023 and used for experiments 4 or 5 days after emergence to wait for sexual maturity.

### Sympatric species of French Guiana

*M. h. helenor* and *M. a. achilles* individuals were wild-caught and raised in insectaries between July and September 2023 in the Amazonian forest of the Kaw Mountains (GPS coordinates: 4.57,–52.21), French Guiana, France.

## Quantification and characterization of coloration and iridescence of *Morpho* butterfly wings

### Measures of wing reflectance

We investigated the variations in iridescence in *M. h. bristowi*, *M. h. theodorus* (from Ecuador), *M. h. helenor* and *M. a. achilles* (from French Guiana), using 10 females and 10 males of each. We estimated the iridescence of the blue patches of the dorsal side of the wings by performing a series of reflectance measurements on wings positioned on a flat surface, following a template to ensure that the wing orientation remained the same across all measurements. Reflectance spectra captured at different angles were recorded on the right forewing, at a specific point located in the wing zone defined by the M3 and Cu1 veins according to the nymphalid ground plan (see *Martin and Reed, 2010*). Wing reflectance was measured using a spectrophotometer (AvaSpec-ULS2048CL-EVO-RS 200–1100 nm, Avantes) coupled with a deuterium halogen light source (AvaLight-DH-S-BAL, Avantes) for 21 combinations of illumination and observation angles, covering the proximo-distal and antero-posterior planes of the wing (*Figure 2A*, see Appendix 1 for extended methods).

These measurements were repeated three times for all individuals to estimate the repeatability of the reflectance measurements at each angle using the R package *rptR* 0.9.22 (*Stoffel et al., 2017*). Since no difference was detected between replicates for any of the tested angles (*Appendix 4—table 1*), the mean of these 3 measurements at each angle for each individual was used in the subsequent analyses.

## Statistical analysis of iridescence

Each illumination/observation position produced a reflectance spectrum whose characteristics (i.e. distribution of reflectance) define the optical properties of the illuminated surface. We focused our analyses on the wavelength range [300–700 nm], relevant for both butterfly (*Briscoe, 2008*) and bird observers (*Bennett and Théry, 2007*). All reflectance spectra were analyzed using the R package *pavo2* 2.9.0 (*Maia et al., 2019*). The statistical analysis of both colorimetric variables and global iridescence was performed using the iridescence data of 10 males and 10 females from each population (*M. h. bristowi*, *M. h. theodorus*, *M. h. helenor,* and *M. a. achilles*).

## Extraction and analysis of colorimetric variables

To identify the main optical effect produced by the wing at each angle, we extracted three commonly used colorimetric variables from each spectrum, using the *summary()* function in *pavo2*: namely Hue (color in the common meaning of the term - measured as the wavelength at the maximum reflectance), Brightness (average proportion of reflected light - measured by the mean reflectance over

each spectra) and Chroma (also referred to as saturation, it characterizes the color purity - measured as the ratio of the reflectance range of each spectra by the brightness). For each individual, we thus extracted these 3 variables for each of the 21 spectra obtained for different combinations of illumination/reflectance angles. Because the normality and heteroscedasticity assumptions were not met for the distributions of values in these three parameters, we relied on permutation-based ANOVAs (*vegan 2.6.4* package, *Oksanen et al., 2001*) to test the effect of sex, angle of measurement, locality, and species on these three colorimetric variables. To specifically test whether differences in iridescence patterns were observed among species or localities, we estimated the interaction between the effects of *Morpho* populations and of the angle of illumination.

## Global iridescence analysis

As iridescence broadly refers to any change in reflectance in relation to illumination and observation angle (*Doucet and Meadows, 2009*), we characterized the iridescence of each individual using all complete spectra obtained at all angular positions (400 wavelengths x 21 angular positions). We applied a Principal Component Analysis (PCA) to this dataset, where each dot corresponds to the iridescence of an individual wing and thereby shows the similarities and differences in iridescence among groups. The samples from Ecuador and French Guiana were analyzed separately to test for more precise differences in iridescence between allopatric and sympatric *Morpho* butterfly populations. Additionally, those comparisons correspond to the phenotypes used in the mate choice experiments (see below). The coordinates of the 10 first PCs (representing 98% of the explained variance) were then extracted and a PERMANOVA was performed to test for the effect of sex, locality, and species on iridescence patterns.

## Modeling how *Morpho* butterflies and avian predators perceive *M. helenor*

To investigate whether the variations observed among the different spectra could be perceived by different observers, we then used the R package *pavo2* 2.9.0 (*Maia et al., 2019*) to model both avian and butterfly visual capabilities. We used the avian UV-sensitive vision model implemented in *pavo* ('*avg.uv*') as a proxy for predator vision and specifically built a model of *M. helenor* vision, using the spectral sensitivity measured in *Pirih et al., 2022*. We built a hexachromatic visual model using the *sensmodel* function, using 345, 446, 495, 508, 570, and 602 nm as the maximum of absorbance in our six simulated cones, respectively. The *vismodel* function was used to compute the quantum catch of each receptor in the *Morpho* vision model under standard daylight illumination for each spectrum. Finally, assuming that the discrimination of colors is limited by photoreceptor noise (Receptor-Noise Limited model, *Vorobyev et al., 2001*), we estimated the distance between colored stimuli as perceived by Morphos using the *bootcoldist* function. We used the default options (weber = 0.1, *weber.achro*=0.1) and set the photoreceptor densities for each cone used in the visual model to 1. The resulting chromatic distances ($dS$), achromatic distances (dL), and confidence intervals were then used to test how much the visual capacities of *Morpho* butterflies could allow the discrimination of wing reflectance (brightness for achromatic distances, hue for chromatic distances) depending on sex, localities, and species, at each angle. Similarly, we used the avian UV-sensitive visual model implemented in *pavo* to test the discrimination of *Morpho* wing reflectance by predators. We used dS ≥1 *Just Noticeable Difference* as the threshold of discrimination.

## Mating experiments

All mate choice experiments took place in cages of similar dimensions (3m x 3m x 2 m), in which *Morpho* butterflies, especially males, were able to fly freely. All *Morpho* butterflies were fed 30 min before the beginning of the experiments taking place between 8 am and 12 am.

### Testing for mate discrimination within *M. helenor*

To study the discrimination of mating partners within the species *M. helenor*, tetrad experiments were performed between *M. h. bristowi* and *M. h. theodorus*. For each experiment, one virgin female and one mature male of both *M. h. bristowi* and *M. h. theodorus* were placed within the same cage. Their starting positions in each corner of the cage were randomized. The first mating occurring out of the four possibilities was recorded, and the experiment was stopped as soon as the male joined their

| Experiments (Location) | Dummy female 1 | Dummy female 2 | Live males | Test |
|---|---|---|---|---|
| 1 (STRI) | *M. h. theodorus* | *M. h. bristowi* | *M. h. theodorus* (n = 33) | Assortative preference within species |
| | | | *M. h. bristowi* (n = 28) | |
| 2 (STRI) | *M. h. bristowi* | *M. h. bristowi* with thinner blue band | *M. h. bristowi* (n = 28) | Discrimination on the pattern |
| 3 (STRI) | *M. h. theodorus* | *M. h. bristowi* with thinner blue band | *M. h. theodorus* (n = 33) | Discrimination on the iridescent blue |
| 4 (French Guiana) | *M. h. helenor* | *M. a. achilles* | *M. h. helenor* (n = 28) | Assortative preference between species |
| | | | *M. a. achilles* (n=37) | |

**Figure 6.** Experimental design of the four male mate choice experiments. Experimental design of the male choice experiments performed between the Morphos originating from Ecuadorian populations of *M. helenor* (eastern population: *M. h. theodorus* and western population: *M. h. bristowi*, experiments 1–3) and from French Guiana (*M. h. helenor* and *M. a. achilles*, experiment 4), along with the associated preferences and visual cues tested. Note that the same males were used for experiments 1, 2, and 3.

genitalia. Experiments could also be ended when no mating occurred within 5 hr of trial. We carried out N=33 trials to estimate whether the first mating observed preferentially involved assortative pairs, due to either male and/or female mate preference, of which N=30 trials actually resulted in assortative or disassortative mating. We statistically tested for departure from the random mating expectation using a Fisher's exact test.

Note that *M. achilles* is not commercially bred and wild females are usually all mated, preventing experiments on virgin females in this species, and thus restricting experiments on virgin females in the commercially-bred species *M. helenor*.

## Identifying specific visual cues used during mate choice by *M. helenor* males

As male *Morpho* tend to patrol in the wild to find females, and because reports show that males are attracted to bright blue coloration in the wild (**Le Roy et al., 2021a**), we specifically tested for male mate choice using visual cues. Females are also likely to be sensitive to color pattern in mate recognition. However, we did not test for female choice because their behavior was not compatible with experiments involving flying around butterfly dummies: female individuals tended to hide away from the males in our experimental cages. To investigate the visual preference of *M. helenor* males for the dorsal color pattern of females, we performed several choice experiments using dummy butterflies built with actual female wings. To ensure that preferences were not triggered by olfactory cues, all female wings used to build the artificial dummies were washed with hexane before any experiment. The details of the experimental design and specific visual cues tested are described in **Figure 6**.

First, we tested male preference for the visual appearance of *M. h. bristowi* and *M. h. theodorus* females (Experiment 1). Since *M. h. bristowi* and *M. h. theodorus* females differ both in terms of color patterns (width of blue band) and iridescence, we then conducted experiments to isolate these two components.

We specifically tested the effect of the proportion of blue to black areas (Experiment 2) by presenting males *M. h. bristowi* to female dummies displaying *M. h. bristowi* wings, except one was modified with black ink to resemble the narrow blue band on *M. h. theodorus* female wings.

We finally tested the effect of the blue iridescent band within the same black and blue pattern (Experiment 3) by presenting *M. h. theodorus* males to a *M. h. theodorus* female with a narrow blue band, and to the modified *M. h. bristowi* female dummy with a narrow band. The wing pattern modifications were performed using a black permanent marker.

To ensure that ink odor would not bias the behavior of *Morpho* males, all dummies used in experiments 2 and 3 contained traces of black marker (either on the modified bands or on the ventral side of the wings of natural females). Nevertheless, we cannot entirely discard the specific visual effect of the marker on the dorsal face.

In all experiments, two female dummies were hanging from the ceiling of the cage, 1.5 m apart, and presented to a live male. The male was released in the experimental cage in front of the two female dummies and was free to interact with them for 10 min. The positions of the female models were swapped 5 min into each trial to avoid potential position-related bias. Two types of male behaviors were recorded. Each time a male flew within a 40 cm radius from a female model, an 'approach' was recorded. Each time a male touched a female dummy, a 'touch' was recorded. The number of 'approaches' and 'touches' toward each female dummy per trial was used to determine male preference. Each male performed this experiment three times on 3 different days to account for potential individual variations in mate preference.

### Testing whether visual cues can act as a pre-zygotic barrier between *M. h. helenor* and *M. a. achilles*

In order to investigate the importance of color vision in species recognition, we then performed a male choice experiment with *M. h. helenor* and *M. a. achilles* (experiment 4 in *Figure 6*) in French Guiana, using natural wings of females *M. h. helenor* and *M. a. achilles* as artificial butterflies following the exact same protocol above.

## Statistical analyses
### Comparing male mating motivation
Because we noticed behavioral differences between the species and subspecies of *Morpho* during the course of the experiments 1–4, we attempted to account for variations in male mating motivation. The approach and touch rates of males, that is, their willingness to approach and touch (respectively) any female dummies during each trial, were defined as the proportion of trials where the males approached and touched (respectively) either female dummies at least once. The approach rate and touch rate of *M. h. bristowi, M. h. theodorus, M. h. helenor,* and *M. a. achilles* males were calculated, and a binomial generalized linear model (GLMM) was used to test whether the approach or touch rate was significantly different between males from different populations.

All statistical analyses were performed with the R package *glmmTMB* (**Brooks et al., 2017**). The normality, homogeneity of variance, and overdispersion tests for the GLMMs were performed using the R package *DHARMa* (**Hartig, 2022**). We controlled for potential individual variations by using the male ID and trial number as a random effect in all the tests.

### Visually-based male mate preferences
We then tested for male preferential attraction for different dummy females. For each male individual, we recorded the number of approaches and touches directed toward the two female dummies and modeled this as a two-column response variable. Similar to the protocol above, we used a binomial GLMM to test whether individuals preferentially interacted with dummy 1 vs. dummy 2, and added a random effect for individual identity to account for repeated observations from the same individual. A significant signal was detected when the probability of touching a dummy was different from the 0.5 proportion expected under random mating and interpreted as a signal of preference.

## Pheromone analysis for sympatric species
### Chemical analyses
We investigated variations in chemical compounds between the two sympatric species *M. helenor* and *M. achilles* sampled in French Guiana. The male and female genitalia of wild-caught *M. h. helenor* and *M. a. achilles* individuals (from 9 up to 12 per sex per species) were dissected and extracted in 20 µL of

dichloromethane, containing 5 µL of octadecane as an internal standard. Two sets of chromatography were performed in order to extract either the light compounds (all volatile carbonate compounds below C16) or the heavier ones (all carbonate compounds above C16) found on the genitalia of those Morphos.

The chemical compounds of those samples were analyzed by gas chromatography and time-of-flight mass spectrophotometry (GC-ToF-MS) using a 8890 GC system (Agilent, Santa Clara, USA) coupled to a PEGASUS BT mass spectrophotometer (LECO, Saint Joseph, USA). The connected capillary column was an Rxi-5ms (30 m x 0.25 mm, df = 0.25 µm, Restek, Bellefonte USA) and helium was used as carrier gas at 1 mL min⁻¹ constant flow rate. 1 µL of each sample was injected in a GC split/splitless injector at 300 °C. Two temperature programs were used: (i) the analysis of the C8-C16 compounds started at 40 °C maintained for 1 min, then increased at 210 °C at 6 °C min⁻¹ rate, finally increased to 340 °C at a 10 °C min⁻¹, and (ii) the analysis of the C16-C30 compounds started at 70 °C maintained for 1 min, then increased to 150 °C at 30 °C min⁻¹ rate and finally increased to 340 °C at a 6 °C min⁻¹ rate. Mass spectra were recorded with a 70 eV electron impact energy. Mixtures of alkanes C8 to C19 and C16 to C44 (ASTM D5442 C16-C44 Qualitative Retention Time Mix, Supelco, Bellefonte, USA) were injected under the same conditions to be used as external standards.

The chromatograms were converted to netCDF format and operated using *MZmine 2.43* for compound detection and area integration. To this aim, chromatograms were cropped before 4 min and after 27 min and baseline-corrected using the Rolling ball algorithm (**Kneen and Annegarn, 1996**). Mass detection was performed using the centroid mass detector, and chromatograms of each mass were constructed using the ADAP algorithm (**Myers et al., 2017**; **Pluskal et al., 2010**). Chromatograms were deconvoluted into individual peaks using the local minimum search algorithm and aligned on a master peak list with the RANSAC algorithm. Finally, the Peak Finder gap filling algorithm was used to search for missing peaks, and a matrix compiling the area under each peak (i.e. the concentration of each component) detected for each butterfly sample was extracted. This matrix was purged from siloxane-like compounds as well as compounds present in less than 10% of the samples. The areas of the blanks were subtracted from the samples, and the final concentration data was used for statistical analysis. Mass spectra were then visualized using the ChromaTOF software, compared to MZmine-selected ones, in order to manually annotate the chemical compounds. Annotations were based on scan comparison with the NIST2017 database, and linear retention indexes (LRI) calculated from the alkane mixtures with free database (https://pubchem.ncbi.nlm.nih.gov/).

## Statistical analyses of the chemical compounds

In order to quantify variations in chemical compounds detected in the different sample, we performed a nonmetric multidimensional scaling (NMDS) ordination using the Bray-Curtis dissimilarity calculation on the concentrations extracted from the chromatograms. By considering the amount of each compound found in our samples as a variable, we performed a permutation MANOVA with the *adonis2()* function found in the *vegan 2* package on R. We tested whether the sex or the species had a significant effect on the chemical bouquet produced by their genitalia.

We also performed an Indicator Value Analysis to identify the chemical compounds significantly associated with our different *Morpho* samples with the *mutlipatt* function from the *indicspecies* package in R using the IndVal index from **Dufrêne and Legendre, 1997**.

## Acknowledgements

The authors would like to thank Aurélie Tournié and Doris Gomez for the advice provided on iridescence measurements. We are also grateful to Owen McMillan from the Smithsonian Tropical Research Institute (Panama) and Mathieu Chouteau from LEEISA in French Guiana (France) for providing facilities to perform the behavioral experiments. We exported the wings of the butterflies from Panama using exportation permit number PA-01-ARB-028–2023, and declared to the French authorities the exportation of butterfly wings from French Guiana to the French metropole. JL PhD was funded by an IBEES grant from Sorbonne Université. This study was funded by the European Union (ERC-2022-COG - OUTOFTHEBLUE - 101088089). Views and opinions expressed are however those of the authors only and do not necessarily reflect those of the European Union or the European Research Council. Neither the European Union nor the granting authority can be held responsible for them.

# Additional information

## Funding

| Funder | Grant reference number | Author |
|---|---|---|
| European Research Council | ERC-2022-COG - OUTOFTHEBLUE - 101088089 | Joséphine Ledamoisel<br>Bruno Buatois<br>Rémi Mauxion<br>Christine Andraud<br>Melanie McClure<br>Vincent Debat<br>Violaine Llaurens |
| Sorbonne Université | IBEES | Joséphine Ledamoisel |

The funders had no role in study design, data collection and interpretation, or the decision to submit the work for publication.

## Author contributions

Joséphine Ledamoisel, Conceptualization, Resources, Formal analysis, Investigation, Visualization, Methodology, Writing - original draft, Writing – review and editing; Bruno Buatois, Methodology, Writing – review and editing; Rémi Mauxion, Christine Andraud, Resources, Methodology, Writing – review and editing; Melanie McClure, Resources, Formal analysis, Methodology, Writing – review and editing; Vincent Debat, Conceptualization, Supervision, Validation, Methodology, Writing – review and editing; Violaine Llaurens, Conceptualization, Supervision, Funding acquisition, Validation, Project administration, Writing – review and editing

## Author ORCIDs

Joséphine Ledamoisel  https://orcid.org/0000-0001-7885-3300
Bruno Buatois  https://orcid.org/0000-0003-3574-8425
Christine Andraud  https://orcid.org/0000-0002-3112-9363
Melanie McClure  https://orcid.org/0000-0003-3590-4002
Vincent Debat  https://orcid.org/0000-0003-0040-1181
Violaine Llaurens  https://orcid.org/0000-0003-1962-7391

Joint Public Review: https://doi.org/10.7554/eLife.106098.3.sa1
Author response https://doi.org/10.7554/eLife.106098.3.sa2

# Additional files

## Supplementary files

MDAR checklist

## Data availability

All raw measurements of wing reflectance and R scripts required to generate the results are publicly available under the DOI 10.5281/zenodo.14389631, or on GitHub.

The following dataset was generated:

| Author(s) | Year | Dataset title | Dataset URL | Database and Identifier |
|---|---|---|---|---|
| Ledamoisel J | 2025 | Wing reflectance of 2 populations of sympatric and allopatric Morpho butterfly sister-species | https://zenodo.org/records/14389631 | Zenodo, 10.5281/zenodo.14388104 |

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

# Appendix 1

## *Morpho* wing reflectance measurement protocols

The reflectance of the right anterior wings of 80 *Morpho* individuals was measured at different angles of illumination using a spectrometer (AvaSpec-ULS2048CL-EVO-RS, Avantes) coupled with a deuterium halogen light source (AvaLight-DH-S-BAL, Avantes) and two optical fibers (FCR-7UVIR200-2−1.5X100 and FC-UVIR200-2−1.5X100, Avantes) supported by an AFH-15 Angled Fibre Holder (Avantes) (***Appendix 1—figure 1***).

First, we measured the specular reflectance. Specularity refers to the reflectance in the direction symmetrical to the illumination relative to the normal of a surface (i.e. the vector perpendicular to the surface of an object): it is the expected direction of maximal reflectance of a perfect mirror. (***Appendix 1—figure 2***). We used a total of 13 specular combinations to quantify the hue variation within samples due to iridescence. First, we positioned the two fibers vertically, that is their common axis was perpendicular to the surface of the wings (forming a 0° angle) to measure the wing reflectance at the normal of the wing. Then, we jointly tilted the two fibers symmetrically to the normal in the proximo-distal plane and measured the reflectance of the wings at (i) a 15° angle from the normal, (ii) a 30° angle from the normal, and (iii) a 45° angle from the normal. The illumination and observation fibers were then switched and the three specular measures were repeated under a new illumination side, allowing the quantification of the specular reflectance of the proximo-distal plane at six different angles of illumination in total. The same protocol was applied by positioning the two fibers symmetrically to the normal in the antero-posterior plane and measuring the reflectance of the wings at six different angles of illumination as well.

We then quantified the variation of brightness of the wings using a 'tilt set-up' (***Appendix 1—figure 3***; ***Gruson et al., 2019***).

We symmetrically positioned the two fibers 30° to the normal of the wings, forming a constant angular span of 60° between the two fibers in the proximo-distal plane of the wings. While keeping this fixed position between the two fibers, we measured the wing reflectance when tilting the fibers (i) 15° toward the normal of the wings and (ii) 15° away from the normal of the wings. The illumination and observation fibers were then switched and two additional measurements were performed on the proximo-distal plane of the wings. The same protocol was applied by positioning the two fibers in the antero-posterior plane and measuring the reflectance of the wings at four new different angles of illumination. This protocol allowed for the quantification of eight new reflectance spectra quantifying iridescence for each sample.

Note that for both methods of measurement, we measured a black and a white reference every time the angle of illumination changed.

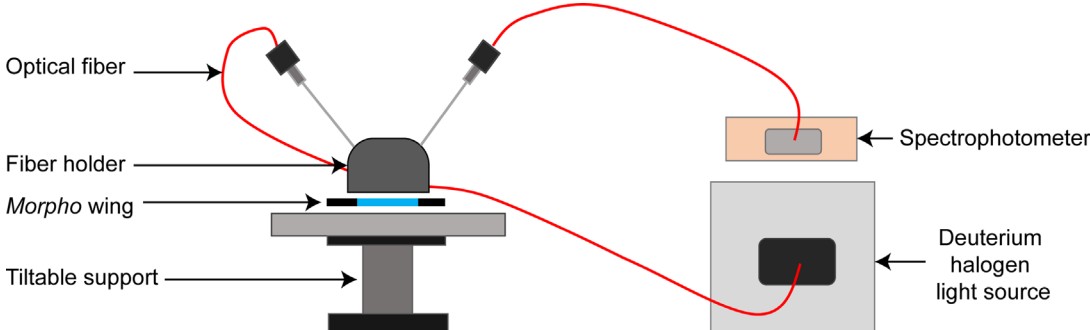

**Appendix 1—figure 1.** Illustration of the set-up used to measure the reflectance of the wings of *Morpho* butterflies at different angles of illumination and observation. We used a fiber holder to precisely control the angle between the two fibers. We ensured the measured surface was flat with a support holding the wings.

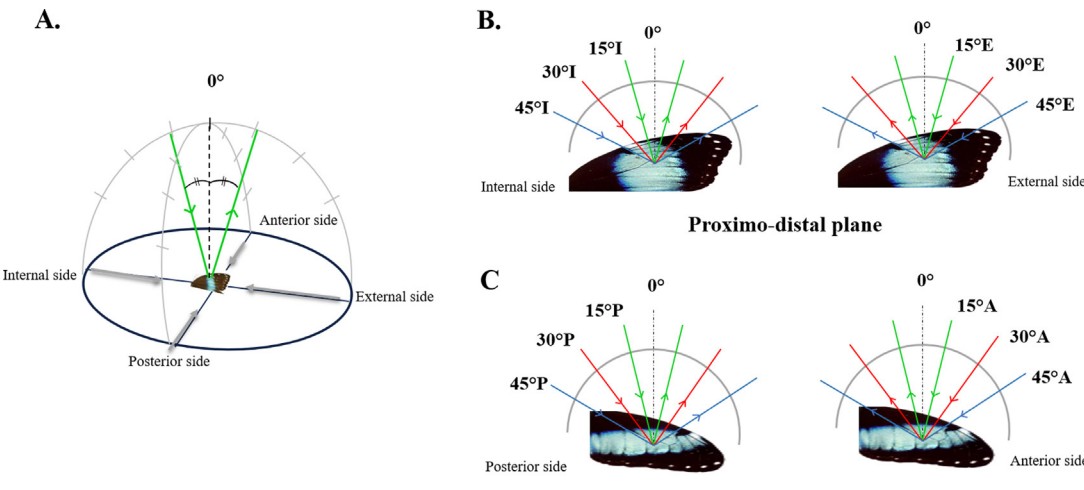

**Appendix 1—figure 2.** Scheme of the specular angles of illumination used to measure the reflectance of the wings. (**A**) shows the coverage of the 13 angles of illumination measured for each wing. The angles analyzed on the proximo-distal plane are represented in (**B**), and the angles analyzed on the antero-posterior plane are shown in (**C**). Each letter used to describe the angles (**I, E, P, A**) refers to the side the light was directed to (Internal, External, Posterior, and Anterior side of wings, respectively).

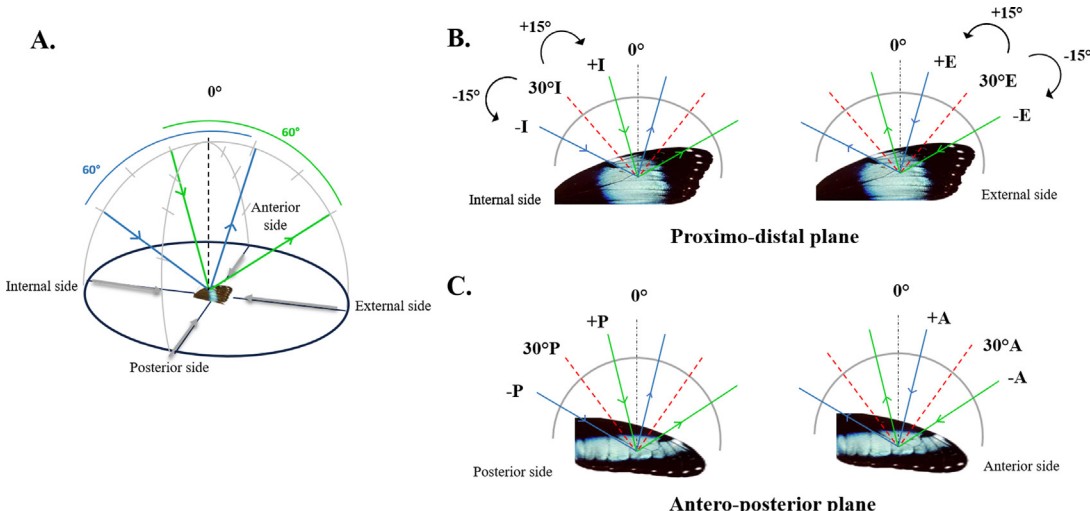

**Appendix 1—figure 3.** Scheme of the angles measured in the tilt set-up. The red dotted lines represent the specular 30° angle measured in the 'specular' set-up. The angular span between the two fibers is kept the same and the two fibers are tilted toward the normal of the wings (annotated with +) or away from it (annotated with -), allowing the additional measurement of the wing's reflectance. This operation is repeated on the Internal, External, Anterior, and Posterior sides (I, E, A, and P, respectively) of the wings.

# Appendix 2

## Statistical results for Figure 2

**Appendix 2—table 1.** Permutation-based ANOVAs performed in order to test whether sex, taxa, or the angle of illumination have an effect on the estimated hue and brightness measured on the proximo-distal plane of *Morpho* wings.

We observe that the effect of sex is always significant on the variations of brightness hinting at sexual dimorphism. Hue is also significantly different between the allopatric sub-species of *M. helenor*, *M. h. theodorus,* and *M. h. bristowi*, whereas it was very similar for the two sympatric species *M. h. helenor* and *M. a. achilles*, suggesting convergence in wing coloration between the two sympatric species.

| | Hue Specular French Guiana | Hue Specular Ecuador | Brightness Specular French Guiana | Brightness Specular Ecuador | Brightness 30° tilt French Guiana | Brightness 30° tilt Ecuador |
|---|---|---|---|---|---|---|
| Sex | 0.062 | 0.001 *** | 0.002 ** | 0.001 *** | 0.001 *** | 0.001 *** |
| Species | 0.188 | 0.001 *** | 0.001 *** | 0.066 | 0.506 | 0.007 ** |
| Angle | 0.001 *** | 0.001 *** | 0.001 *** | 0.001 *** | 0.001 *** | 0.001 *** |
| Sex:Species | 0.030 * | 0.002 ** | 0.001 *** | 0.001 *** | 0.020 * | 0.114 |
| Species:Angle | 0.485 | 0.638 | 0.188 | 0.362 | 0.006 ** | 0.050 * |

# Appendix 3

## Annotation of the chemical compounds significantly associated to both sexes of *M. helenor* or *M. achilles* using an indicator value analysis.

**Appendix 3—table 1.** Annotation of the chemical compounds significantly associated with both sexes of *M. helenor* or *M. achilles* using an Indicator value analysis.

We used an Indicator Value analysis to find the chemical compounds allowing us to discriminate the males and females of *M. helenor* and *M. achilles* separately. Because we used a protocol allowing for the detection of small peaks during the MZmine analysis to not exclude 'pheromone-like' molecules, our analysis is sensitive to small spectral variations. This could explain why some compounds were associated with different (but similar) MZmine-detected occurrences. X refers to an undetermined double bond position or configuration (*Z* or *E*). The 'Ions' column describes the spectrum with the major ion, followed by other ions in intensity order, and the underlined visible ion corresponding to the molecular ion.

| Sex | Dataset | Annotation | LRI | Ions | Species | p-value |
|---|---|---|---|---|---|---|
| Males | C8 to C16 | (*E*)-*β*-Ocimene | 1048 | 93; 79; 41; 136 | *M. helenor* | 0.0491 |
| | | Phenyl ethyl alcohol | 1112 | 91; 122; 65 | *M. achilles* | 0.0004 |
| | | Unknown compound 1 | 1123 | 80; 52; 73; 98; 124 | *M. helenor* | 0.0311 |
| | | Ethyl octanoate | 1195 | 88; 57; 101; 43; 73; 127; 172 | *M. helenor* | 0.0001 |
| | | Tetradec-1-ene | 1390 | 43; 55; 69; 83; 97; 196 | *M. achilles* | 0.0282 |
| | | Ethyl decanoate | 1394 | 88; 101; 43; 73; 155; 200 | *M. helenor* | 0.0001 |
| | | Dodec-x-enol | 1454 | 55; 68; 41; 82; 96; 184 | *M. helenor* | 0.0002 |
| | | Ethyl dodecanoate | 1593 | 88; 101; 43; 73; 155; 228 | *M. helenor* | 0.0001 |
| | | (x)-Ethyl Tetradec-x-enoate | 1764 | 88; 55; 41; 96; 166; 254 | *M. helenor* | 0.0001 |
| | | (x)-Tetradec-x-enyl acetate | 1784 | 43; 68; 54; 82; 96; 194 | *M. helenor* | 0.0001 |
| | C16 to C30 | Ethyl hexadecanoate | 1993 | 88; 101; 43; 73; 155; 284 | *M. helenor* | 0.0001 |
| | | Unknown compound 2 | 2046 | 44; 207; 49; 83; 55; 69 | *M. helenor* | 0.0001 |
| | | Geranyl decanoate | 2148 | 69; 93; 41; 121; 136; 308 | *M. helenor* | 0.0001 |
| | | (*Z*)-Ethyl Octadec-9-enoate | 2168 | 55; 41; 69; 88; 83; 310 | *M. helenor* | 0.0001 |
| | | Ethyl octadecanoate | 2193 | 88; 101; 43; 55; 157; 312 | *M. helenor* | 0.0001 |
| | | Unknown compound 3 | 3109 | 44; 207; 57; 71; 85; 281 | *M. achilles* | 0.0096 |
| | | (x)-Tetradec-x-enyl hexadecanoate | 3148 | 68; 82; 96; 194; 43; 450 | *M. helenor* | 0.0001 |
| | | Unknown compound 4 | 3342 | 207; 43; 55; 73; 81; 95 | *M. achilles* | 0.0159 |
| | | (x)-Tetradec-x-enyl octadecanoate | 3350 | 68; 82; 96; 194; 43; 57 | *M. helenor* | 0.0001 |
| Females | C8 to C16 | Dec-1-ene | 990 | 56; 41; 70; 83; 97; 140 | *M. helenor* | 0.028 |
| | | Dodec-1-ene | 1190 | 43; 55; 69; 83; 97; 168 | *M. helenor* | 0.0302 |

# Appendix 4

## Repeatability of the measurements of iridescence

**Appendix 4—table 1.** Repeatability (R) of the measurements of iridescence performed on the wings of *M. h. bristowi*, *M. h. theodorus*, *M. h. helenor*, and *M. a. achilles*.

The repeatability was measured for the values of Brightness, Hue, and Chroma at every angle of illumination.

| R | SE | Emp.2.5% | Emp.97.5% | p-value | Variable | Angle_ID |
|---|---|---|---|---|---|---|
| 0.93195387 | 0.01307307 | 0.90333517 | 0.95236442 | 4.38E-77 | Chroma | 0 |
| 0.8815925 | 0.02194644 | 0.82807312 | 0.91642739 | 2.02E-58 | Brightness | 0 |
| 0.93731446 | 0.01264365 | 0.90649166 | 0.95552368 | 7.07E-80 | Hue | 0 |
| 0.81478231 | 0.03228571 | 0.73922202 | 0.86625459 | 1.06E-43 | Chroma | 15 A |
| 0.67046172 | 0.05265538 | 0.54844062 | 0.75544574 | 1.20E-25 | Brightness | 15 A |
| 0.91021841 | 0.01686965 | 0.87067891 | 0.93479852 | 1.06E-67 | Hue | 15 A |
| 0.77559976 | 0.03806848 | 0.68813618 | 0.83531032 | 1.52E-37 | Chroma | 15E |
| 0.6708999 | 0.05014185 | 0.56249193 | 0.74961219 | 1.10E-25 | Brightness | 15E |
| 0.87101943 | 0.02490225 | 0.81147963 | 0.90678958 | 1.41E-55 | Hue | 15E |
| 0.83143394 | 0.02937952 | 0.76803356 | 0.87725532 | 9.15E-47 | Chroma | 15I |
| 0.73512952 | 0.04070802 | 0.63907404 | 0.8005203 | 2.51E-32 | Brightness | 15I |
| 0.91839526 | 0.01578924 | 0.88039618 | 0.94433766 | 6.33E-71 | Hue | 15I |
| 0.79982088 | 0.0350237 | 0.7162314 | 0.85456168 | 3.39E-41 | Chroma | 15 P |
| 0.7198958 | 0.04867878 | 0.61651339 | 0.79526635 | 1.36E-30 | Brightness | 15 P |
| 0.88930422 | 0.02020534 | 0.8443417 | 0.92331442 | 1.14E-60 | Hue | 15 P |
| 0.79421599 | 0.03598134 | 0.7191763 | 0.8551282 | 2.62E-40 | Chroma | 30 A |
| 0.67732938 | 0.04919037 | 0.56751756 | 0.76016964 | 2.80E-26 | Brightness | 30 A |
| 0.9289188 | 0.01305597 | 0.89974257 | 0.95162324 | 1.33E-75 | Hue | 30 A |
| 0.70444585 | 0.04668047 | 0.60669801 | 0.78777167 | 6.07E-29 | Chroma | 30E |
| 0.71178183 | 0.045488 | 0.61393897 | 0.79154588 | 1.03E-29 | Brightness | 30E |
| 0.87874635 | 0.02331122 | 0.82296662 | 0.91409894 | 1.25E-57 | Hue | 30E |
| 0.6550905 | 0.05264239 | 0.53816018 | 0.74507705 | 2.76E-24 | Chroma | 30I |
| 0.74554083 | 0.04261923 | 0.64566456 | 0.81391148 | 1.41E-33 | Brightness | 30I |
| 0.87582059 | 0.02347792 | 0.82228053 | 0.91484364 | 7.75E-57 | Hue | 30I |
| 0.9041886 | 0.01878598 | 0.85773967 | 0.93360667 | 1.64E-65 | Chroma | 30 P |
| 0.81205069 | 0.03280043 | 0.74302844 | 0.86991471 | 3.15E-43 | Brightness | 30 P |
| 0.97112453 | 0.00585086 | 0.95746447 | 0.9795759 | 1.88E-106 | Hue | 30 P |
| 0.85400673 | 0.0260558 | 0.79757803 | 0.89737906 | 1.77E-51 | Chroma | 45 A |
| 0.77039882 | 0.0374088 | 0.69002671 | 0.83570177 | 8.13E-37 | Brightness | 45 A |
| 0.90203576 | 0.01936148 | 0.85524126 | 0.93076324 | 9.14E-65 | Hue | 45 A |
| 0.75913134 | 0.03786569 | 0.67916231 | 0.82351654 | 2.66E-35 | Chroma | 45E |
| 0.81757629 | 0.02896174 | 0.75431007 | 0.87015713 | 3.40E-44 | Brightness | 45E |
| 0.82071206 | 0.03200376 | 0.74873736 | 0.86827257 | 9.31E-45 | Hue | 45E |

*Appendix 4—table 1 Continued on next page*

*Appendix 4—table 1 Continued*

| R | SE | Emp.2.5% | Emp.97.5% | p-value | Variable | Angle_ID |
|---|---|---|---|---|---|---|
| 0.74259191 | 0.04170908 | 0.64865071 | 0.80893911 | 3.23E-33 | Chroma | 45I |
| 0.71182855 | 0.04478549 | 0.62281411 | 0.7850276 | 1.02E-29 | Brightness | 45I |
| 0.84104241 | 0.02886878 | 0.77925018 | 0.88941637 | 1.10E-48 | Hue | 45I |
| 0.83345745 | 0.03061064 | 0.76221461 | 0.88316779 | 3.69E-47 | Chroma | 45 P |
| 0.72576245 | 0.0435171 | 0.62298039 | 0.79267106 | 3.01E-31 | Brightness | 45 P |
| 0.79537205 | 0.03348855 | 0.72196461 | 0.85099538 | 1.73E-40 | Hue | 45 P |
| 0.76096223 | 0.03875962 | 0.67466542 | 0.82646812 | 1.53E-35 | Chroma | 30A+ |
| 0.57007786 | 0.05936361 | 0.44073031 | 0.68032181 | 6.32E-18 | Brightness | 30A+ |
| 0.82901303 | 0.03012831 | 0.75897509 | 0.87566813 | 2.67E-46 | Hue | 30A+ |
| 0.89830354 | 0.01859355 | 0.85566402 | 0.9260342 | 1.65E-63 | Chroma | 30E+ |
| 0.60590647 | 0.05531483 | 0.48615964 | 0.6922734 | 2.16E-20 | Brightness | 30E+ |
| 0.86040869 | 0.02652343 | 0.79782212 | 0.90284859 | 5.85E-53 | Hue | 30E+ |
| 0.83151877 | 0.03001095 | 0.76702206 | 0.87982723 | 8.82E-47 | Chroma | 30I+ |
| 0.80264773 | 0.03329181 | 0.72607819 | 0.85682716 | 1.18E-41 | Brightness | 30I+ |
| 0.75740508 | 0.03975858 | 0.67291885 | 0.81765625 | 4.46E-35 | Hue | 30I+ |
| 0.83149921 | 0.03123307 | 0.76071749 | 0.88420299 | 8.89E-47 | Chroma | 30P+ |
| 0.7315642 | 0.0418415 | 0.64205063 | 0.80411359 | 6.54E-32 | Brightness | 30P+ |
| 0.54180507 | 0.06347357 | 0.39773273 | 0.64264721 | 3.63E-16 | Hue | 30P+ |
| 0.82564933 | 0.03141051 | 0.75785206 | 0.87820262 | 1.15E-45 | Chroma | 30A- |
| 0.70304331 | 0.04565709 | 0.609797 | 0.78119994 | 8.47E-29 | Brightness | 30A- |
| 0.70430795 | 0.04743961 | 0.59725879 | 0.77909707 | 6.27E-29 | Hue | 30A- |
| 0.76953439 | 0.04063247 | 0.68300252 | 0.83811216 | 1.07E-36 | Chroma | 30E- |
| 0.84800758 | 0.02819233 | 0.78156911 | 0.88972866 | 3.73E-50 | Brightness | 30E- |
| 0.84527977 | 0.0268152 | 0.7854896 | 0.88880349 | 1.43E-49 | Hue | 30E- |
| 0.9142409 | 0.01566538 | 0.87865615 | 0.93819666 | 3.01E-69 | Chroma | 30I- |
| 0.83297497 | 0.02802545 | 0.76998544 | 0.88057836 | 4.59E-47 | Brightness | 30I- |
| 0.88792846 | 0.02111732 | 0.84024728 | 0.92021421 | 2.95E-60 | Hue | 30I- |
| 0.7596208 | 0.04114026 | 0.6698537 | 0.826702 | 2.29E-35 | Chroma | 30P- |
| 0.7115446 | 0.04568159 | 0.61202912 | 0.78354985 | 1.09E-29 | Brightness | 30P- |
| 0.9412519 | 0.01089593 | 0.91657877 | 0.95702933 | 4.35E-82 | Hue | 30P- |

