## [Editor Report · eLife Assessment]

This study presents a **valuable** assessment of and **solid** evidence for increased similarity in visual appearance combined with increased chemical differences between two butterfly species in sympatry compared with differences between three populations of one of the two species in allopatry. The similarity in visual appearance hints to an evolutionary response to shared predators (but alternative explanations are possible). Thus, the difference in chemical signaling likely helps to avoid between-species mating in sympatry.

---

## [Referee Report · Joint Public Review]

Summary:

Ledamoisel et al. examined the evolution of visual and chemical signals in closely related Morpho butterfly species to understand their role in species coexistence. Using an integrative, state-of-the-art approach combining spectrophotometry, visual modeling, and behavioral mate choice experiments, they quantified differences in wing iridescence and assessed its influence on mate preference in allopatry and sympatry. They also performed chemical analyses to determine whether sympatric species exhibit divergent chemical cues that may facilitate species recognition and mate discrimination. The authors found iridescent coloration to be similar in sympatric Morpho species. Furthermore, male mate choice experiments revealed that in sympatry, males fail to discriminate conspecific females based on coloration, reinforcing the idea that visual signal convergence is primarily driven by predation pressure. In contrast, the divergence of chemical signals among sympatric species suggests their potential role in facilitating species recognition and mate discrimination. The authors conclude that interactions between ecological pressures and signal evolution may shape species coexistence.

Strengths:

The study is well-designed and integrates multiple methodological approaches to provide a thorough assessment of signal evolution in the studied species. We appreciate the authors' careful consideration of multiple selective pressures and their combined influence on signal divergence and convergence. Additionally, the inclusion of both visual and chemical signals adds an interesting and valuable dimension to the study, enhancing its importance. Beyond butterflies, this research broadens our understanding of multimodal communication and signal evolution in the context of species coexistence.

Reviewing Editor comment:

The authors have improved their submission after revisions and responded to the previous concerns of the reviewers.

---

## [Author Response]

The following is the authors’ response to the original reviews.

**Reviewer #1 (Public review):**
Summary:In this study, Ledamoisel et al. examined the evolution of visual and chemical signals in closely related Morpho butterfly species to understand their role in species coexistence. Using an integrative, state-of-the-art approach combining spectrophotometry, visual modeling, and behavioral mate choice experiments, they quantified differences in wing iridescence and assessed its influence on mate preference in allopatry and sympatry. They also performed chemical analyses to determine whether sympatric species exhibit divergent chemical cues that may facilitate species recognition and mate discrimination. The authors found iridescent coloration to be similar in sympatric Morpho species. Furthermore, male mate choice experiments revealed that in sympatry, males fail to discriminate conspecific females based on coloration, reinforcing the idea that visual signal convergence is primarily driven by predation pressure. In contrast, the divergence of chemical signals among sympatric species suggests their potential role in facilitating species recognition and mate discrimination. The authors conclude that interactions between ecological pressures and signal evolution may shape species coexistence.Strengths:The study is well-designed and integrates multiple methodological approaches to provide a thorough assessment of signal evolution in the studied species. I appreciate the authors' careful consideration of multiple selective pressures and their combined influence on signal divergence and convergence. Additionally, the inclusion of both visual and chemical signals adds an interesting and valuable dimension to the study, enhancing its importance. Beyond butterflies, this research broadens our understanding of multimodal communication and signal evolution in the context of species coexistence.Weaknesses:(1) The broader significance of the findings needs to be better articulated. While the authors emphasize that comparing adaptive traits in sympatry and allopatry provides insights into selective processes shaping reproductive isolation and coexistence, it is unclear what key conceptual or theoretical questions are being addressed. Are these patterns expected under certain evolutionary scenarios? Have they been empirically demonstrated in other systems? The authors should explicitly state the overarching research question, incorporate some predictions, and better contextualize their findings within the existing literature. If the results challenge or support previous work, that should be highlighted to strengthen the study's importance in a broader context.

We thank the reviewer for their valuable feedback. We understand that the framing of the results and the discussion may fail to convey the broader significance of our findings. In the first version of the manuscript, we framed our manuscript around the processes shaping reproductive isolation and co-existence in sympatry, but now realize that this question was too broad in regards to our results. We thus strictly focused on outlining the importance of ecological interactions in the evolution of traits in sympatric species. In the revised version of the manuscript, we rewrote the first paragraph of the introduction to introduce context regarding the effect of ecological interactions on trait evolution (lines 43-60). We then explicitly introduce the theoretical question investigated in our paper (i.e. “we investigate how ecological interactions in sympatry can constrain natural and sexual selection shaping trait evolution”, lines 62-63) and our predictions regarding the evolution of traits in sympatry vs. allopatry (lines 74-80). We also added predictions regarding our experiments on Morpho at the end of the introduction (lines 146-157). As a result, the discussion is now better aligned with the introduction, by discussing the putative effect of predation and mate choice on the evolution of wing iridescence in Morpho.

(2) The motivation for studying visual signals and mate choice in allopatric populations (i.e., at the intraspecific level) is not well articulated, leaving their role in the broader narrative unclear. In particular, the rationale behind experiments 1, 2, and 3 is not well defined, as the authors have not made a strong case for the need for these intraspecific comparisons in the introduction. This issue is further compounded by the authors' primary focus on signal evolution in sympatry throughout both the results and the discussion. For instance, the divergence of iridescence in allopatry is a potentially interesting result. But the authors have not discussed its implications.

We now clearly state in the introduction our motivation for studying visual signals and mate choice in allopatric populations (lines 74-80, lines 146-157). We argued that intraspecific comparisons help identify whether visual cues can be used in mate recognition between phylogenetically close subspecies, between whom visual resemblance is supposed to be higher than between closely-related species (tetrad experiment, and experiment 1). As M. h. bristowi and M. h. theodorus have different wing pattern, we also used this comparison to identify the traits involved in male mate preference within a species, testing the importance of iridescent color (experiment 2) or iridescent patterning (experiment 3). The results of those experiments can then be used to assess whether these traits are used in species recognition between sympatric species. See also our answers to recommendations 11 and 15 from reviewer #1.

Overall, given that the primary conclusions are based on results and analyses in sympatry, the role of allopatric populations in shaping these conclusions needs to be better integrated and justified. Without a stronger link between the comparative framework and the study's key takeaways, the use of allopatric populations feels somewhat peripheral rather than central to the study's aim. Since the primary conclusions remain valid even without the allopatric comparisons, their inclusion requires a clearer rationale.

To make a stronger case for the use of the allopatric population in our manuscript, we strengthened the justification behind the study of intraspecific allopatric populations vs. interspecific sympatric populations, as the iridescence measurements and the mate choice experiments in allopatric populations can serve as a baseline in studying how species interactions can shape the evolution of traits and mate recognition when compared to sympatric populations. Following your major comment #1, we rewrote the introduction to include a justification to the need for studying allopatric vs. sympatric populations (lines 74-80), and also further highlighted the need to study iridescence in sympatric species to fully understand the trait evolution of sympatric species in the discussion (339-343).

(3) While the authors demonstrate that iridescence is indistinguishable to predators in sympatry, they overstate the role of predation in driving convergence. The present study does not experimentally demonstrate that iridescence in this species has a confusion effect or contributes to evasive mimicry. Alternatively, convergence could result from other selective forces, such as signal efficacy due to environmental conditions, rather than being solely driven by predation.

We acknowledge that our study does not directly demonstrate that iridescence contributes to evasive mimicry. We did tone down the interpretation of the results in the discussion and state that predation is not the only selective pressure that could have promoted a convergent evolution of iridescence in sympatric species, as iridescence is a trait that could be involved in thermoregulation (lines 346-353) and camouflage (lines 363-369) for example. We made sure to mention that convergence in iridescent signals in sympatry is only an indirect support to the evasive mimicry hypothesis, and that further research is still needed, including direct predation experiments, to show that this convergence is indeed triggered by predation (lines 391-396).

**Reviewer #2 (Public review):**
This study presents an investigation of the visual and chemical properties and mating behaviour in Morpho butterflies, aimed at addressing the nature of divergence between closely related species in sympatry. The study species consists of three subspecies of Morpho helenor (bristowi, theodorus, and helenor), and the conspecific Morpho achilles achilles. The authors postulate that whereas the iridescent blue signals of all (sub)species should function as a predator reduction signal (similar to aposematism) and therefore exhibit convergence, the same signals should indicate divergence if used as a mating signal, particularly in sympatric populations. They also assess chemical profiles among the species to assess the potential utility of scent in mediating species/sex discrimination.The authors first used reflectance spectrometry to calculate hue, brightness, and chroma, plus two measures of "iridescence" (perhaps better phrased as angular dependence) in each (sub)species. This indicated the ubiquitous presence of sexual dimorphism in brightness (males brighter), which also appears to be the case for iridescence (Figure 3A-B). Analysis of these data also indicated that whereas there is evidence for divergence among subspecies in allopatry, the same evidence is lacking for species in sympatry (P = 0.084). This was supported further by visual modelling, which showed that both conspecifics and birds should be (theoretically) capable of perceiving the colour difference among allopatric populations of M. helenor, whereas the same is not true for the sympatric species.The authors then conducted mate choice trials, first using live individuals and second using female dummies. The live experiments indicated the presence of assortative mating among the two subspecies of M. helenor (bristowi and theodorus). The dummy presentations indicated (a) bristowi males prefer conspecific wings, whereas theodorus have no preference, (b) bristowi males prefer the con(sub)specific colour pattern, (c) theodorus prefer the con(sub)specific iridescence when the pattern is manipulated to be similar among female dummies. A fourth experiment, using sympatric M. achilles and M. helenor, indicated no preference for conspecific female dummies. Finally, chemical analysis indicated substantial differences between these two species in putative pheromone compounds, and especially so in the males.The authors conclude that the similarity of iridescence among species in sympatry is suggestive of convergence upon a common anti-predation signal. Despite some behavioural evidence in favourof colour (iridescence)-based mate discrimination, chemical differences between Achilles and Helenor are posed as more likely to function for species isolation than visual differences.Overall, I enjoyed reading this manuscript, which presents a valiant attempt at studying visual, chemical and behavioural divergence in this iconic group of butterflies.Major commentsMy only major comment concerns the authors' favoured explanation for aposematism (or evasive mimicry) for convergence among species, which is based upon the you-can't-catch-me hypothesis first presented by Young 1971. Although there is supporting work showing that iridescent-like stimuli are more difficult to precisely localize by a range of viewers, most of the evidence as applied to the Morpho system is circumstantial, and I'm not certain that there is widespread acceptance of this hypothesis. Given that the present study deals with closely-related (sub)species, one alternative explanation - a "null" hypothesis of sorts - is for a lack of divergence (from a common starting point) as opposed to evolutionary convergence per se. in other words, two subspecies are likely to retain ancestral character states unless there is selection that causes them to diverge. I feel that the manuscript would benefit from a discussion of this alternative, if not others. Signalling to predators could very well be involved in constraining the extent of convergence, but this seems a little premature to state as an up-front conclusion of this work. There is also the result of a *dorsal* wing manipulation by Vieira-Silva et al. 2024 which seems difficult to reconcile in light of this explanation. Whereas this paper is cited by the authors, a more nuanced discussion of their experimental results would seem appropriate here.

We thank the reviewer for their constructive comments on our manuscript. We appreciate the reviewer’s concern regarding the way iridescence convergence between sympatric species is discussed in our manuscript, which align with similar concerns raised by Reviewer 1. Indeed, the you-can't-catch-me hypothesis has not been yet empirically tested in Morpho, this is currently a working hypothesis only supported by indirect lines of evidence.

Among the 30 known Morpho species, iridescence is most likely the ancestral character, notably because iridescence is a trait shared by a majority of Morpho (we now mention this in the introduction lines 108-110). In this paper, we thus did not aim to identify the evolutionary forces involved in the appearance of iridescence in this group, but rather wanted to understand to what extent ecological interactions can impact the diversification (or not) of this trait. As such, the dorsal manipulations performed in Vieira-Silva et al 2024 showing that iridescence in Morpho may have a similar effect than crypsis does not impact our working hypothesis. Instead, we use VieraSilva et al 2024 to discuss the potential anti-predator effect of iridescence, that could potentially promote convergent evolution of iridescent patterns.

In the main text, we now clearly mention our null hypothesis: under a scenario of neutral evolution of iridescence, we would expect that the divergence in wing coloration between two M. helenor subspecies would be lower than between two different Morpho species (M. helenor and M. achilles) and showed that our results sharply differ from this null expectation.

We then improved the discussion by adding alternative hypotheses potentially explaining the convergent iridescent signal detected in sympatric species: we discussed the expected effect under neutral evolution (lines 339-343), but also added alternative hypotheses regarding the diversification of iridescence due to camouflage (lines 363-369), predator evasion (lines 373-377) and thermoregulation (lines 346-353).

**Reviewer #3 (Public review):**
The authors investigated differences in iridescence wing colouration of allopatric (geographically separated) and sympatric (coexisting) Morpho butterfly (sub)species. Their aim was to assess if iridescence wing colouration of Morpho (sub)species converged or diverged depending on coexistence and if iridescence wing colouration was involved in mating behaviour and reproductive isolation. The authors hypothesize that iridescence wing colouration of different (sub)species should converge in sympatry and diverge in allopatry. In sympatry, iridescence wing colouration can act as an effective antipredator defence with shared benefits if multiple (sub)species share the same colouration. However, shared wing colouration can have potential costs in terms of reproductive interference since wing colouration is often involved in mate recognition. If the benefits of a shared antipredator defence outweigh the costs of reproductive interference, iridescence wing colouration will show convergence and alternative mate recognition strategies might evolve, such as chemical mate recognition. In allopatry, iridescence wing colouration is expected to diverge due to adaptation to different local conditions and no alternative mate recognition is expected.Strengths:(1) Using allopatric and sympatric (sub)species that are closely related is a powerful way to test evolutionary hypotheses(2) By clearly defining iridescence and measuring colour spectra from a variety of angles, applying different methods, a very comprehensive dataset of iridescence wing colouration is achieved.(3) By experimentally manipulating wing coloration patterns, the authors show visual mate recognition for M. h. bristowi and could, in theory, separate different visual aspects of colouration (patterns VS iridescence strength).(4) Measurements of chemical profiles to investigate alternative mate recognition strategies in case of convergence of visual signals.Weaknesses:In my opinion, studies should be judged on the methods and data included, and not on additional measurements that could have been taken or additional treatments/species that should be included, since in most ecological and evolutionary studies, more measurements or treatments/species can always be included. However, studies do need to ensure appropriate replication and appropriate measurements to test their hypothesis AND support their conclusions. The current study failed to ensure appropriate replication, and in various cases, the results do not support the conclusions.First, when using allopatric and sympatric (sub)species pairs to test evolutionary hypotheses, replication is important. Ideally, multiple allopatric and sympatric (sub)species pairs are compared to avoid outlier (sub)species or pairs that lead to biased conclusions. Unfortunately, the current study compares 1 allopatric and 1 sympatric (sub)species pair, hence having poor (no) replication on the level of allopatric and sympatric (sub)species pairs,

We would like to thank the reviewer for their constructive feedback. We agree that replication is important to test evolutionary hypotheses and that our study lacks replication for allopatric and sympatric Morpho populations. Ideally, one would require several allopatric and sympatric replicates to conclude on the effect of species interaction in trait evolution. Our study is a preliminary attempt at answering this question, covering a few Morpho populations but proposing a broad assessment of iridescence and mate preference for those populations. We clearly mentioned in the discussion that investigating multiple populations is needed to test whether the trend we observed in this paper can be generalized (line 388-392).

Second, chemical profiles were only measured for sympatric species and not for allopatric (sub)species, which limits the interpretation of this data. The allopatric (sub)species could have been measured as non-coexistence "control". If coexistence and convergence in wing colouration drives the evolution of alternative mate recognition signals, such alternative signals should not evolve/diverge for allopatric (sub)species where wing colouration is still a reliable mate recognition cue. More importantly, no details are provided on the quantification of butterfly chemical profiles, which is essential to understand such data. It is unclear how the chemical profiles were quantified and what data (concentrations, ratios, proportions) were used to perform NDMS and generate Figure 5 and the associated statistical tests.

We recognize that having the chemical profiles of the genitalia of the Morpho from the allopatric populations would have made a stronger case in favor of reinforcement acting on the divergence of the chemical compounds found on the genitalia of the sympatric Morpho species. Due to limited access to the biological material needed at the time of the chromatography, we could not test for lower divergence in the chemical profiles of allopatric Morpho butterflies. We made sure to mention this limitation in the discussion (lines 457-461).

We already stated in the methods that we compiled the area under the peak of each components found in the chromatograms of our samples and that we performed all the statistical analyses on this dataset. To make it clearer, we mention in the new version of the manuscript that the area under the peak of each component allows to measure the concentration of the components (in the methods lines 720, 723, 733). We also added some precisions in the legend of Figure 5.

Third, throughout the discussion, the authors mention that their results support natural selection by predators on iridescent wing colouration, without measuring natural selection by predators or any other measure related to predation. It is unclear by what predators any of the butterfly species are predated on at this point

We made sure to mention in the introduction (line 132-136) and in the discussion (line 373-377) that previous predation experiments performed on Morpho and other butterflies showed evidence that birds are likely predators for these species. These observations lead us to test for the putative effect of predation on the evolution of their color pattern, without directly testing predatory rates. We made sure this information is transparent in the revised manuscript, and now precise that assessing wing convergence is only an indirect way of testing the escape mimicry hypothesis (line 393-396).

To continue on the interpretation of the data related to selection on specific traits by specific selection agents: This study did not measure any form of selection or any selection agent. Hence, it is not known if iridescent wing colouration is actually under selection by predators and/or mates, if maybe other selection agents are involved or if these traits converge due to genetic correlations with other traits under selection. For example, Iridescent colouration in ground beetles has functions as antipredator defence but also thermo- and water regulation. None of these issues are recognized or discussed.

The lack of discussion of alternative selective pressures involved in the evolution of iridescence was pointed out by all reviewers. We thus modified the text to account for this comment, and no longer limit our discussion to the putative effects of predation. We now specifically discuss alternative hypotheses, including crypsis (362-369) and thermoregulation (line 346-353).

Finally, some of the results are weakly supported by statistics or questionable methodology.Most notably, the perception of the iridescence coloration of allopatric subspecies by bird visual systems. Although for females, means and errors (not indicated what exactly, SD, SE or CI) are clearly above the 1 JND line, for males, means are only slightly above this line and errors or CIs clearly overlap with the 1 JND line. Since there is no additional statistical support, higher means but overlap of SD, SE or CI with the baseline provides weak statistical support for differences.

We thank the reviewer for bringing interpretation issues concerning the chromatic distances of allopatric Morpho species measured with a bird vision model. We made sure to be nuanced in the description of this graph in the results section (line 208-212). Note that this addition does not change our main conclusion stating that Morpho and predator visual models better discriminate iridescence differences between allopatric subspecies than between sympatric species.

We now also clearly mention in the figure’s legend that the error bars represent the confidence intervals obtained after performing a bootstrap analysis, in addition to the mention of the nature of the error bars already mentioned in the methods (line 580).

Regarding the assortative mating experiment, the results are clearly driven by M. bristowi. For M. theodorus, females mate equally often with conspecifics (6 times) as with M. bristowi (5 times). For males, the ratio is slightly better (6 vs 3), but with such low numbers, I doubt this is statistically testable. Overall low mating for M. bristowi could indicate suboptimal experimental conditions, and hence results should be interpreted with care.

We recognize that the tetrad experiment results are mainly driven by M. bristowi’s behavior as already mentioned in the results (line 231-232) but we now also mention it in the discussion (lines 401-402). This experiment would have benefited from more replicates, but the limited access to live males and virgin females for both subspecies was a limiting factor. Fisher’s exact test used to assess assortative mating is specifically appropriate to small sample sizes. We recognize that the sampling size is not ideal, however it is still statistically testable.

Regarding the wing manipulation experiment, M. theodorus does not show a preference when dummies with non-modified wings are presented and prefers non-modified dummies over modified dummies. This is acknowledged by the authors but not further discussed. Certainly, some control treatment for wing modification could have been added.

The use of controls to consider the effect of wing modification and odor by the permanent marker were already mentioned in the methods (lines 636-639). Following your recommendation and comments from the other reviewers, we now mention the use of this control in the results (lines 278283). We also address a potential issue that would have resulted in the rejection of these modified dummies by live males: we cannot be sure whether butterflies perceive these modifications as equivalent to natural coloration (lines 281-282). An additional control could have been used, adding black ink on the black dorsal parts of the pattern to assess its potential visual effect. The constraints on sampling unfortunately did not allow to add another treatment.

Overall, the fact that certain measurements only provide evidence for 1 of the 2 (sub)species (assortative mating, wing manipulation) or one sex of one of the species (bird visual systems) means overall interpretation and overgeneralization of the results to both allopatric or sympatric species should be done with care, and such nuances should ideally be discussed.The aim of the authors, "to investigate the antagonistic effects of selective pressures generated by mate recognition and shared predation" has not been achieved, and the conclusions regarding this aim are not supported by the results. Nevertheless, the iridescence colour measurements are solid, and some of the behavioural experiments and chemical profile measurements seem to yield interesting results. The study would benefit from less overinterpretation of the results in the framework of predation and more careful consideration of methodological difficulties, statistical insecurities, and nuances in the results.

Overall, we would like to thank all reviewers for their thorough assessment of our work. We understand that the imbalance between mate choice data, visual model data and chemical data only gives us a partial assessment of species recognition in Morpho butterflies, thus requiring more precision in the interpretation and the discussion of our results. We made sure to add balanced interpretations in our discussion, by mentioning the lack of replicates for allopatric and sympatric populations (lines 391-392), and the lack of chemical characterization of allopatric species (lines 458361, see previous comments) and by being more transparent on methodological limitations that we failed to convey in the first version of our manuscript. We brought nuance to our discussion and also discussed alternative hypotheses to predation to explain the convergence of iridescence found in sympatry.

Reviewing Editor Comments:While all reviewers acknowledge the value of your data, they converge in their recommendations to tone down the evolutionary interpretations. Ideally, to test your main hypothesis, you would need several species pairs, or if only one, as in your case, replicated sympatric and allopatric sites for both species. Furthermore, your more specific hypotheses about convergence (vs. nondivergence), response to predators (vs. other environmental variables), and avoiding interspecific mating in sympatry (vs. not avoiding it in allopatry) would require appropriate alternative treatments/controls. We therefore recommend that you focus on those statements that you can support with your experiments and data, and introduce these statements in the introduction with reference to the appropriate literature.
**Reviewer #1 (Recommendations for the authors):**
(1) Line 25: This stated aim seems a bit off. The authors did not sensu stricto quantify 'how shared adaptive traits may shape genetic divergence' in this study. I suggest rewriting or deleting this whole sentence altogether. The study's aim is already clear in lines 29-34.

We deleted the mention of the characterization of genetic divergence, since this study did not focus on any genetic analysis.

(2) Line 34: The authors here state that they compared allopatric vs sympatric populations. This is strictly not true for M. Achilles. Further, the results after this sentence focus solely ondivergence/convergence in sympatry, nothing at the intraspecific level and implications of the findings

We now mention that we tested allopatric vs. sympatric species of M. helenor only (lines 28-29). We also mention that the behavioral experiments were based on intraspecific comparisons, and discuss the implications of this result in the discussion.

(3) Line 35: 'convergence driven by predation': this is a strong statement and cannot be directly inferred from the present set of experiments. Consider toning it down.

We added nuance to this statement by rephrasing it “suggesting that predation may favors local resemblance” (lines 32-33)

(4) Line 36: Replace 'behavioral results' with 'behavioral experiments' or something similar.

Corrected

(5) Line 45-49: These opening statements need some citations.

We provided references for the first few lines, by citing terHorst et al 2018 (line 44) underlining the importance of species interactions in trait evolution, and Blomberg et al 2003 (line 45) showing that closely-related species tend to resemble each other by quantifying the phylogenetic signal of various traits.

(6) Line 83, 165: 'visual effect', not sure what the authors are referring to. Please rewrite.

We defined “visual effect” as the way wing color patterns could be perceived by predators or mates. We removed mentions of “visual effect” and directly used its definition instead.

(7) Line 105 onwards: This section of the introduction could benefit from more concise writing. The authors might consider reducing the number of specific examples and instead offering broader general statements, supported by citations from multiple studies.

We reduced the number of examples given in this paragraph and used general statements supported by multiple citations as examples. (lines 102-119).

(8) Line 108-110: This sentence seems to be redundant with the previous one.

We merged this sentence with the previous one to improve clarity. (lines 103-105)

(9) Line 140: 'with chemical defenses': include citations here.

We added citations of Joron et al 1999 and Merrill et al 2014, which document the evolution of convergent wing patterns (mimicry) in butterfly species with chemical-defenses.

(10) Line 149: This is a bit of a stretch. Note that genetic divergence could be influenced by many other things, not only the processes that the authors examined.

We agree with the reviewer that the study of the convergent vs. divergent evolution of visual cues is not enough to fully understand the mechanisms allowing genetic divergence between species. Because this paper does not focus on characterizing genetic divergence, we removed it from the manuscript to avoid oversimplification.

(11) Line 151: Again. Here, the author's primary focus seems to be at an interspecific level. One is left to wonder about the need for comparisons at the intraspecific level in M.helenor and the implications. Please clarify

In the end of the introduction (lines 146-157), we specifically highlighted the importance of intraspecific comparisons. While studying the effect of sympatry on the evolution of the iridescent color pattern, we use this intraspecific comparison as a baseline to account for convergence or divergence of iridescence in a sympatric interspecific pair of Morpho, because under neutral evolution two subspecies are expected to be more similar than two different species (this assumption has been clarified line 147-148). We also used intraspecific mate choice to test for the use of visual cues in mate recognition (experiment 1) and to test what type of signal could be perceived by Morphos (the iridescent coloration or the iridescent pattern, experiment 2 and 3). These results help contextualize the interspecific mate choice, focused on determining whether visual cues could also be used in species recognition. Since we show that iridescent coloration is important in mate recognition at the intraspecific scale, it helps understand why species recognition is low at the interspecific scale because of wing color convergence between M. helenor and M. achilles.

(12) Line 154: 'signals on mate preferences'.

Corrected.

(13) Line 189: 'At the intraspecific level', maybe in the brackets include 'allopatric populations' just so the results are in a similar format as in the color contrast section below.

We added details to make clearer that the intraspecific level is studied between allopatric Morpho populations (line 189).

(14) Line 189-192: Please rearrange the figure (current B as A and vice versa) or present the results in order as in the figure (interspecific first and then intraspecific level).

We rearranged Figure 3 so that the intraspecific comparison (allopatric population) appears as A and the interspecific level (sympatric population) appears as B, to follow the order of presentation in the main text.

(15) Line 232: The motivation behind experiments 1, 2, and 3 is unclear. The authors have not made a strong point in the introduction about the need for these comparisons at an intraspecific level. Given that the authors are focused on divergence/convergence at an interspecific level, this set of experiments seems to be irrelevant to the present study. The implications of these findings are also not discussed.

We added motivation to the use of experiment 1, 2, and 3 in the introduction (lines 151-154) by stating that those experiments were used to assess whether blue color could indeed be used as a mating cue in Morpho helenor (experiment 1) and to try to understand what part of the visual signal is important in mate choice in Morpho helenor: the wing pattern (experiment 2) or the iridescent coloration (experiment 3). Although motivation for these experiments was not detailed in our manuscript, we already discussed the implications of the results of experiments 1, 2 and 3 in the discussion by stating that visual cues can take many forms and that considering both color AND pattern is important in understanding visual cues (lines 408-416). We carefully reworked this new version to make it more straightforward.

(16) Line 260: Insert 'wild-type' before model to ensure similar wording as in the previous section.

Corrected.

(17) Line 286: Insert 'sympatric' after mimetic.

Corrected.

(18) Line 307: Include a reference to the figures or table where these results are presented.

We now mention in the main text that the different proportions of beta-ocimene found between males M. helenor and M. achilles are shown in Table S2.

(19) Line 343: These inferences are speculative. Add a line here, something like 'although this warrants further research in this species'.

We detailed what additional experiments are needed lines 388-396.

(20) Line 357: The authors have not discussed their results on iridescence divergence in allopatric populations (line 190) and its implications.

We now made clear in the beginning of the discussion that the divergence of iridescence in allopatric populations is used as a baseline to test for convergent iridescence between species (lines 339-343).

(21) Line 361 onwards: This first paragraph is a bit confusing, as the results mainly focus on allopatry, while the title refers to sympatry.

To avoid confusion between the title and the content of the discussion, we divided the last part of the discussion into two different parts. As the first paragraph mainly focus on allopatry, we isolated it and titled it “Iridescent color patterns can be used as mate recognition cues in M. helenor” (line 498). The next paragraph of the discussion, focusing on the sympatric Morpho populations, has been titled “Evolution of visual and olfactory cues in mimetic sister-species living in sympatry” (line 418).

(21) Line 383: visual cues 'as' poor species.

Corrected.

(23) Line 405: Why females here and not males? This is again confusing since the authors tested for male mate choice in the main experiments. Some background information on sex-specific mate choice in the methods might help.

In this specific sentence, we talk about performing mate choice experiments to test for the discrimination of olfactory cues by females (and not males) because we found a high divergence in the chemical compounds found on male genitalia. Although female chemical compounds could also be used as a cue by males in mate recognition, olfactive mate choice is often driven by female choice in butterflies. We recognize that this perspective does not line up with the mate choice presented in our results section which focused on male mate choice based on visual cues, because of ecological reasons (Morpho males tend to be attracted to bright blue colorations but not females) and technical reasons (in cages, females tend to hide away from the males or male dummies, and this behavior is not compatible with experiments involving flying around false males). In the discussion, we made sure to precise that the perspective we cite here is about testing the implications of divergence in male olfactory cues (line 454). We also added motivation to why we chose to investigate male (and not female) mate choice based on visual cues in the methods (lines 613-618) and in the results (219-223).

(24) Line 417: This inference is speculative. Consider toning it down.

We rewrote the sentence: “We find evidence of converging iridescent patterns in sympatry suggesting that predation could play a major role in the evolution of iridescence. Further work is nevertheless needed to directly test this hypothesis and establish the important of evasive mimicry in Morpho” (lines 465-468).

(25) Line 429: 'Convergent trait evolution leads to mutualistic interactions enhancing coexistence'. Careful here. It is not very evident how convergent trait evolution (iridescence) is mutualistic in this case, as there is no experimental evidence for evasive mimicry yet. Consider rewording or toning this sentence down.

We agree with the reviewer and removed this statement, only keeping the end of the sentence: “Altogether, this study addresses how convergence in one trait as a result of biotic interactions may alter selection on traits in other sensory modalities, resulting in a complex mosaic of biodiversity. (lines 479-481).

(26) Line 442: Since the samples come from a breeding farm, I have a few questions. How are the authors sure about the location where the specimens were collected? How long have they been kept in captivity? Have they been subjected to any artificial selection? More details are needed here.

Since M. helenor bristowi and M. helenor theodorus are only found in the wild in West and East Ecuador respectively, those M. helenor subspecies can only be collected in those two allopatric populations. Their phenotype is directly linked to their geographic repartition, this is how we made sure about their collect location. M. h. theodorus we used in this study were caught in East Ecuador in Tena, and M. h. bristowi were caught in West Ecuador in Pedro Vincente Madonado. We received pupae from the breeding farm, meaning that the Morpho used for the experiments were raised in captivity since their date of emergence. Upon emergence, they were transferred into cages for 4 to 5 days to wait for sexual maturity before performing the tetrad and mate choice experiments. This information was added to the method (lines 490-496).

(27) Line 476: Include some citations supporting this statement.

We now cite Bennett and Théry (2007), reviewing avian color vision, and Briscoe (2008), characterizing the sensitivity of the photoreceptors found in the eyes of butterflies. Both citations show that the 300-700nm range is seen by avian and butterfly visual systems.

(28) Line 480 onwards: Please clarify if the analysis used only one value (mean?) per species, sex, angle of measurement, and locality or included data from multiple individuals.

The analyses of both colorimetric variables and global iridescence were performed using iridescence data from multiple individuals (10 males and 10 females from M. h. bristowi, M. h. theodorus, M. h. helenor and M. a. achilles), for which we measured iridescence at 21 angles of illumination. Sampling size are mentioned lines 507, 515, 540-542.

(29) Line 510: Is there a specific reason that authors did not investigate achromatic contrasts? Provide some justification here. Or include the results of achromatic contrasts in the supplement.

We added the achromatic results in the supplement and in the results (lines 200-204). For both the avian visual model and the Morpho visual model, the confidence intervals always overlapped with the JND threshold, showing that neither birds nor butterflies could theoretically discriminate the wing reflectance brightness in allopatric and sympatric populations.

(30) Line 552 onwards: I may have missed it. It is not entirely clear why the authors focused on male mate choice rather than female preference for visual cues. The authors should explicitly justify this choice and cite previous studies demonstrating that male mate choice, rather than female preference, is important in this species. This should be stated in the results section as well.

We added a paragraph in the method (lines 613-618) to describe the ecological and technical reasons leading to testing only male mate choice using visual cues (also see our response to recommendation #23).

(31) Line 537 onwards: What was the criterion used to score that mating had occurred? Why first mating and not how long they were mating? Please add these details.

We stopped the experiment as soon as a male/female pair was formed by joining their genitalia (we added this information in the method lines 599-600). Since the tetrad experiment involves the interaction of two males and two females from different subspecies, we considered that mate choice happened before the formation of any couple, and is not necessarily dependent on how long they mate by observing their mating behavior. For instance, we witnessed avoidance behaviors from females that systematically hide their genitalia and refused to join their abdomen to some males, while being very ‘open’ to others (but did not quantify it).

(32) Line 571: The authors used a black permanent marker to modify wing patterns but did not validate whether butterflies perceive these modifications as equivalent to natural coloration. It is possible that the alterations introduced unintended visual cues and may explain why most males rejected the dummies (line 267). The authors should acknowledge this limitation here.

We now acknowledge this limitation in the method (lines 638-639) and in the results section (lines 278-283).

(33) Line 591: Insert 'above' after protocol.

Corrected.

(34) Line 605: If the authors included random effects in their model, then it should be generalized linear mixed model (GLMM) and not GLM as they wrote.

We indeed included a random effect in our model accounting for male ID and trial number, we thus replaced “GLM” by “GLMM” in the manuscript.

(35) Line 615: This set of analyses does not seem to account for pseudo-replication, as the data were recorded from the same male more than once (Line 583). Please clarify and redo the analysis with the GLMM framework

We run new analyses using the GLMM framework: we used a binomial GLMM to test whether individuals preferentially interacted with dummy 1 vs. dummy 2 while accounting for pseudoreplication. The previously detected tendencies hold true with these new analyses, except for the visual mate discrimination of M. achilles: we now find statistical evidence that M. achilles tend to approach more their conspecifics during the mate choice experiment, although the signal is weak (line 297-307). Indeed, while we previously concluded that both species in sympatry (M. helenor and M. achilles) could not discriminate their conspecific mates, we now emphasize that M. achilles is somewhat sensitive to some visual signals. However, its estimated probability of approaching a conspecific is only 0.54, which is low compared to the estimated probability of approaching (0.61) or touching (0.84) a con-subspecific for M. bristowi. We thus concluded that even though some visual cues could be relevant for mate recognition, they are less reliable for male choice in sympatric populations were color patterns are more convergent, compared to allopatric populations. We thus updated Figure 4 and Figure S8 and S9, which are now picturing the probability of approaching or touching a conspecific or con-subspecific with the updated pvalues retrieved from the GLMM analyses. We also updated the results (line 297-307) and the discussion (lines 430-438) to bring nuance to our previous results.

(36) Line 963: Figure 3D. Is there a particular reason for comparing allopatric populations only within Ecuador rather than between Ecuador and French Guiana for M. helenor? Please clarify.

We aimed at comparing the putative discrimination of blue coloration using visual models vs. what the butterflies actually discriminate using mate choice experiments. Since we only performed mate choice experiments involving M. h. bristowi x M. h. theodorus (allopatric populations within Ecuador) and M. h. helenor x M. a. achilles (sympatric population from Ecuador), we only looked at those comparisons using visual models. We added this precision lines (559-560).

(37) Line 980: Are these predicted probabilities or just mean proportions as written in line 614? Then the label should be changed to 'Proportion of approaches' or something similar.

Following our answer to recommendation #35, the points now represent the probability of touching a conspecific in the graph for each male, for every trial of every male tested. We corrected the legend of the figure.

**Reviewer #2 (Recommendations for the authors):**
(1) Line 25: "...therefore facilitating co-existence in sympathy".

Corrected.

(2) Line 28: "contrasting" instead of contrasted.

Corrected.

(3) Line 33: begin a new sentence at the colon.

Corrected.

(4) Line 49: the phrase "habitat filtering" is unclear and should perhaps be defined or qualified.

We replaced “habitat filtering” by its definition and cited Keddy (1992), describing the community assembly rules and defining habitat filtering (line 46)

(5) Line 52: remove "even".

Corrected.

(6) Line 53: divergent suites may also result because traits are often constrained by genetic architecture (multivariate genetic covariances). This is discussed at length and specifically in relation to ornamental coloration by Kemp et al. 2023

We rewrote the introduction and focused on only reviewing the ecological interactions promoting trait divergence in sympatric species, and did not mention genetics in this paper.

(7) Line 87: (and throughout) refer to "colouration" or "colour pattern" rather than "colourations".

Corrected.

(8) Line 151: Remove "To do so,".

Corrected.

(9) Line 191: I would like to see the degrees of freedom for this test.

We added the F-statistic=2.09 and the degrees of freedom df=1 of this test, and for all the following tests.

(10) Line 201: (and throughout) replace "on" with "of".

Corrected.

(11) Line 205: modelling the visual properties of the wings allows one to infer what is theoretically visible/distinguishable. The modelling is useful but not necessarily definitive of vision/behaviour per se under different conditions in the wild. I therefore think it is appropriate to phrase the wording around the modelling approach more carefully. Perhaps refer to "theoretical" or "inferred" discriminability, or state (e.g.) that species should/should not be capable of perceiving differences based on the modelling data. You do this well in your wording of lines 207-209. This need not apply in the discussion because you're then dealing with the combination of modelling results and behaviour (mating trials).

We agree with the reviewer that visual modelling only allows to infer what is theoretically discriminated by the butterflies, and that the wording of our sentence is confusing. We therefore modified the sentence to account for those precisions: “Morpho butterflies and predators can theoretically visually perceive the difference in the blue coloration between different subspecies of M. helenor…… using both bird and Morpho visual models” (line 206-209).

(12) Line 222: Either the chi-square test or Fisher's exact test should be sufficient (why report both?)

Chi-square test relies on large-sample assumptions (expected counts>5) whereas Fischer’s exact test does not and is valid even with small or unbalanced sample sizes. Since the M. bristowi female/M. h. theodorus male paring only occurred 3 times, we do not meet the primary assumptions to apply a Chi-square test, although it is significant. We used a Fischer’s test to confirm the results. Using both and finding that both tests are significant shows that the results are robust, although they may appear redundant. To simplify, we remove the results of the Chisquare test and only keep the Fisher’s test in the methodology and the results.

(13) Line 224 (and throughout): Degrees of freedom should be provided for statistical tests.

We reported the statistic value and the degrees of freedom for all mentions of the statistical tests in the main text, except for the Fischer test which does not rely on an asymptotic distribution like the Chi-squared distribution as it is an exact test.

(14) Lines 266-267: This sentence has interest, but it is rather vague at present. Wouldn't your controls account for the effect of manipulation? This could be explained further.

During our mate choice experiments, all Morpho female dummies used for the experiments were painted with black markers, either on their dorsal blue band to modify their blue iridescent phenotype, or on their ventral side, thus controlling for the effect of manipulation. However, we cannot rule out that the modification of the dorsal blue iridescence could have had a “repulsive” effect for males for several reasons. For example, depending on the visual discrimination of darker colors by Morphos, the painted black band could have a slightly different color compared to the dark “brown” usually surrounding their blue iridescent patterns. We now explain this in the results (lines 278-283) and in the methodology (lines 638-639)

(15) Line 316: I'm not certain that the similarity is best described as "striking", given a P-value of 0.084 for this contrast

We agree with the reviewer and removed this adjective for this line.

(16) Lines 387-390: This sentence is puzzling because, theoretically speaking, we should expect selection on visual preference to be heightened (not relaxed) in sympatry if colouration isincluded among the traits used in mate selection. I'm not certain I have understood the meaning here.

We would like to thank the reviewer for pointing out this typo. If shared predatory pressures favors convergent evolution of color pattern, then the visual signals become less reliable for species recognition. As a result, sexual selection on visual preference is heightened and becomes stronger, favoring the evolution of alternative cues used to discriminate conspecific mates. We changed the sentence and now write “the convergent evolution of iridescent wing patterns… may have negatively impact visual discrimination and favored the evolution of divergent olfactory cues” (lines 457-458).

(17) Line 529: Mating experiments. Given that these are quite large butterflies, I wondered whether a 3x3x2m cage would be sufficient in size to allow the expression of male courtship. A brief description of the courtship behaviour in these species or Morphos generally would be a useful addition to the paper.

A cage this size was enough for the males to express a flight behavior similar to what can be seen in nature, while also being able to see the females (live females or dummies). We tried to perform mate experiments in a larger cage (7m x 5m x 3m) but the trials were not conclusive because male did not find the dummies depending on where they were flying in the cage. A 3mx3mx2m cage is a good compromise maximizing interactions while still allowing enough space to fly. We now describe Morpho male behavior and female behavior in the methods (lines 613-618).

(18) Line 546: Why are both tests needed (chi-square AND Fisher's exact)?

Similarly to our answer on recommendations #12, were used both tests to show robustness in the statistical results. We only kept the Fisher’s test results to simplify the results.